# The choline transporter Slc44a2 controls platelet activation and thrombosis by regulating mitochondrial function

J. Allen Bennett[1], Michael A. Mastrangelo[1], Sara K. Ture[1], Charles O. Smith [2], Shannon G. Loelius[3], Rachel A. Berg[4], Xu Shi[5], Ryan M. Burke[1], Sherry L. Spinelli[4], Scott J. Cameron [1], Thomas E. Carey[6], Paul S. Brookes [2], Robert E. Gerszten[5], Maria Sabater-Lleal [7,8], Paul S. de Vries[9], Jennifer E. Huffman [10], Nicholas L. Smith[11,12], Craig N. Morrell[1] & Charles J. Lowenstein [1✉]

Genetic factors contribute to the risk of thrombotic diseases. Recent genome wide association studies have identified genetic loci including *SLC44A2* which may regulate thrombosis. Here we show that Slc44a2 controls platelet activation and thrombosis by regulating mitochondrial energetics. We find that *Slc44a2* null mice (*Slc44a2(KO)*) have increased bleeding times and delayed thrombosis compared to wild-type (*Slc44a2(WT)*) controls. Platelets from *Slc44a2(KO)* mice have impaired activation in response to thrombin. We discover that Slc44a2 mediates choline transport into mitochondria, where choline metabolism leads to an increase in mitochondrial oxygen consumption and ATP production. Platelets lacking Slc44a2 contain less ATP at rest, release less ATP when activated, and have an activation defect that can be rescued by exogenous ADP. Taken together, our data suggest that mitochondria require choline for maximum function, demonstrate the importance of mitochondrial metabolism to platelet activation, and reveal a mechanism by which Slc44a2 influences thrombosis.

[1] Aab Cardiovascular Research Institute, Department of Medicine, University of Rochester Medical Center, Rochester, NY 14642, USA. [2] Department of Pharmacology and Physiology, University of Rochester Medical Center, Rochester, NY 14642, USA. [3] Department of Microbiology and Immunology, University of Rochester Medical Center, Rochester, NY 14642, USA. [4] Department of Pathology and Laboratory Medicine, University of Rochester Medical Center, Rochester, NY 14642, USA. [5] Division of Cardiovascular Medicine, Beth Israel Deaconess Medical Center, Boston, MA, USA. [6] Department of Otolaryngology-Head and Neck Surgery, University of Michigan, Ann Arbor, MI, USA. [7] Cardiovascular Medicine Unit, Department of Medicine, Karolinska Institutet, Center for Molecular Medicine, Karolinska University Hospital, Stockholm, Sweden. [8] Genomics of Complex Diseases, Research Institute of Hospital de la Santa Creu i Sant Pau (IIB Sant Pau), Barcelona, Spain. [9] Human Genetics Center, Department of Epidemiology, Human Genetics, and Environmental Sciences, School of Public Health, The University of Texas Health Science Center at Houston, Houston, TX, USA. [10] Center for Population Genomics, MAVERIC, VA Boston Healthcare System, Jamaica Plain, MA, USA. [11] Department of Epidemiology, Cardiovascular Health Research Unit, University of Washington, Seattle, WA, USA. [12] Seattle Epidemiologic Research and Information Center, Department of Veterans Affairs Office of Research and Development, Seattle, WA, USA. ✉email: clowens1@jhmi.edu

Venous thromboembolism (VTE) is a disease characterized by thrombosis in veins (deep vein thrombosis (DVT)) or in the lung (pulmonary embolism)[1]. VTE has a strong genetic basis, with the risk of disease increased 2.5-fold if a sibling is affected[2]. However, most of the genetic risk factors for thrombosis are incompletely understood. The INVENT Consortium used GWAS to identify genetic variants associated with VTE in a cohort of over 60,000 human subjects, including a genetic variant located within the gene *SLC44A2* which was associated with a ~20% increased risk of thrombosis in replication and discovery cohorts[3,4].

The biological and physiological roles of the protein SLC44A2 are not well understood[5,6]. The function of SLC44A2 is unknown, but it shares homology with choline transporters such as SLC5A7[7,8]. GWAS studies have associated the *SLC44A2* locus with human phenotypes including: hearing loss, Meniere's disease, and venous thrombosis[3,9]. Recent studies have explored the role of SLC44A2 in thrombosis[10–13]. Two studies found that Slc44a2 promotes thrombosis in a mouse model of laser injury or venous stenosis but did not identify the mechanisms underlying this phenomenon[11,13]. A search for mechanisms of Slc44a2 affecting thrombosis found that Slc44a2 does not affect VWF levels in mice[13]. Another study explored the influence of Slc44a2 upon plasma proteins, and finding no difference in plasma proteins between wild-type and Slc44a2 null mice, concluded that Slc44a2 must influence thrombosis through cellular based mechanisms[12].

We now show that Slc44a2 is a mitochondrial choline transporter that regulates mitochondrial synthesis of ATP, platelet activation and thrombosis.

## Results

**Slc44a2 promotes hemostasis and thrombosis.** We first determined the expression of Slc44a2 using qPCR and immunoblotting in murine and human tissues. Slc44a2 RNA is expressed in all tissues examined (Fig. 1a). Slc44a2 protein was detected in human and murine platelets (Fig. 1b, c). Relative expression of Slc44a2 is higher in the heart than in most other tissues for reasons that are unknown. Mice lacking *Slc44a2*, designated as *Slc44a2(KO)* mice, are global *Slc44a2* null mice that lack Slc44a2 expression in all organs including platelets and bone marrow (Fig. 1c)[8].

SLC44A2 has been genetically linked to thrombosis in humans, so we next characterized the effect of Slc44a2 on hemostasis and thrombosis in mice. We measured the time to hemostasis after tail transection in *Slc44a2(WT)* and *Slc44a2(KO)* mice. *Slc44a2 (KO)* mice have a greatly prolonged bleeding time, up to 50% longer than wild-type mice, suggesting a defect in hemostasis (Fig. 1d). We then used intravital microscopy to measure the time to formation of an occlusive thrombus in mesenteric arteries after FeCl₃ treatment. *Slc44a2(KO)* mice have an increased time to mesenteric artery thrombosis (Fig. 1e). Next we explored the role of Slc44a2 in a murine model of DVT and found that *Slc44a2 (KO)* mice have decreased DVT formation following ligature constriction of the inferior vena cava (IVC) (Fig. 1f–h).

**Slc44a2 in platelets increases hemostasis.** We then explored the effect of Slc44a2 in the bone marrow compartment and in platelets. The bleeding effect of Slc44a2 is dependent on Slc44a2 in bone-marrow-derived cells, since transplantation of bone marrow from *Slc44a2(KO)* mice transfers the bleeding defect to *Slc44a2 (WT)* mice (Fig. 1i). We used ultrasound to measure the time to formation of an occlusive thrombus in the carotid artery after FeCl₃ treatment. Carotid artery thrombosis times are delayed in *Slc44a2(WT)* mice which have a bone marrow compartment

lacking Slc44a2 (Fig. 1j, k). Taken together, our data suggest that Slc44a2 in hematopoietic cells promotes the development of thrombosis and contributes to normal hemostasis.

The bone marrow transplantation experiments suggest but do not prove that Slc44a2 exerts its effects on thrombosis through a cell derived from the bone marrow. In order to determine whether or not this Slc44a2 effect is specific to platelets, we transfused washed platelets from WT and KO donors into KO recipients. *Slc44a2(KO)* mice have shorter bleeding times when receiving platelets from *Slc44a2(WT)* mice instead of from *Slc44a2(KO)* mice (Fig. 1l). (WT platelets improve the bleeding time in *Slc44a2(KO)* mice but do not fully restore the bleeding time to normal wild-type bleeding times, probably because there are still residual KO platelets within the transfused *Slc44a2(KO)* mice.) This suggests that at least part of the prothrombotic effect of Slc44a2 is due to its role in platelets.

**Slc44a2 increases platelet activation.** We hypothesized that Slc44a2 directly regulates platelet activation, since the bleeding phenotype is dependent on the presence of Slc44a2 in bone marrow derived cells. We found that Slc44a2 is expressed in human and murine platelets (Fig. 1b, c). We harvested platelets from *Slc44a2(WT)* and *Slc44a2(KO)* mice, exposed them to vehicle or thrombin, and measured formation of platelet aggregates using light transmission aggregometry. *Slc44a2(KO)* platelets aggregate less than *Slc44a2(WT)* platelets after thrombin treatment (Fig. 2a–d). In particular, the time to 50% maximal aggregation is prolonged, the rate of aggregation is slower, and the percent of aggregation at 3 min is lower in *Slc44a2(KO)* platelets (Fig. 2b–d).

We compared the response of *Slc44a2(WT)* and *Slc44a2(KO)* platelets to various agonists, such as thrombin which signals through G protein coupled receptors, and convulxin which interacts with glycoprotein VI and signals through a (hem) immunoreceptor tyrosine-based activation motif-dependent pathway[14,15]. *Slc44a2(KO)* platelets expose less P-selectin after thrombin stimulation (Fig. 2e). *Slc44a2(KO)* platelets expose less P-selectin than *Slc44a2(WT)* platelets when treated with various agonists (Fig. 2f). *Slc44a2(KO)* platelets also have less activation of GPIIbIIIA than *Slc44a2(WT)* platelets when treated with thrombin (Fig. 2g). However, *Slc44a2(KO)* and *Slc44a2(WT)* platelets have similar levels of dense granule release, measured by CD63 translocation (Fig. 2h). These data suggest that Slc44a2 regulates platelet alpha-granule release and platelet GPIIbIIIa activation.

We next explored the mechanisms through which Slc44a2 regulates platelet activation. We hypothesized that Slc44a2 regulates mitochondrial activity and ATP synthesis. Our hypothesis is based on the observations of others, showing that Slc44a2 is a choline transporter, and mitochondria can use choline to generate betaine, in the process generating reactive oxygen species and ATP[16–18]. To explore the effect of Slc44a2 upon platelet generation of ROS, we compared ROS production from *Slc44a2 (WT)* and *Slc44a2(KO)* platelets after thrombin stimulation. *Slc44a2(KO)* platelets produce less ROS after thrombin activation than *Slc44a2(WT)* platelets (Fig. 2i). We found that mitochondria are the source of the oxidative burst after thrombin stimulation, since inhibitors of mitochondrial enzymes abolish thrombin stimulated ROS (Fig. 2j). We next confirmed that platelet derived mitochondrial ROS mediate platelet activation. Inhibitors of mitochondrial complexes that produce ROS decreased platelet activation (Fig. 2k). These studies suggest that mitochondria are the source of ROS which accompany platelet activation. Taken together, our data suggest that Slc44a2 controls mitochondrial production of ROS during platelet activation.

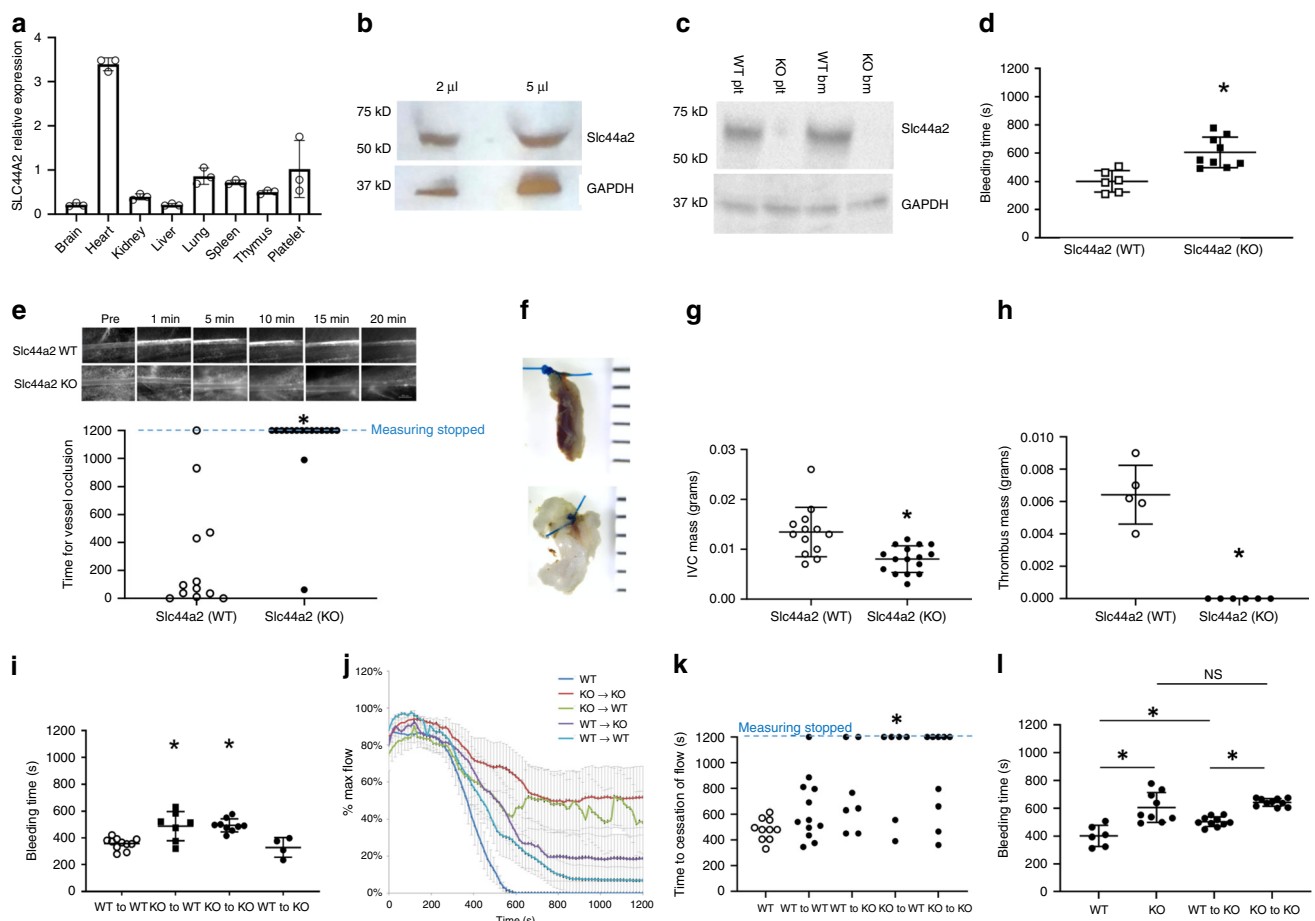

**Fig. 1 Slc44a2 is expressed in platelets and regulates hemostasis and thrombosis in mice. a** RNA levels of Slc44a2 relative to ß-actin in murine organs were measured by qPCR ($n = 3 \pm$ biologically independent samples ±S.D.). **b** Protein levels of SLC44A2 in normal human platelets were measured by immunoblotting. **c** Protein levels of Slc44a2 in mouse platelets and mouse bone marrow were measured by immunoblotting. **d** The bleeding time of Slc44a2(WT) and Slc44a2(KO) mice was measured after tail transection ($n = 6$ WT and 8 KO mice ±S.D. and *$P < 0.01$ in a two-tailed Student's $t$ test). **e** The time for mesenteric arterial thrombosis after FeCl₃ treatment was measured by intravital microscopy. *For WT vs. KO, the Fisher's exact test statistic is 0.0001 and the result is significant at $P < 0.05$. **f** Representative image of inferior vena cava 6 h after IVC constriction, with WT above and KO below. **g** Quantification of IVC mass containing IVC segment and thrombus 6 h after IVC constriction ($n = 13$ WT and 15 KO mice ±S.D. and *$P < 0.01$ in a two-tailed Student's $t$ test). **h** Quantification of thrombus mass isolated from IVC 6 h after IVC constriction ($n = 5$ WT and 6 KO mice ±S.D. and *$P < 0.01$ in a two-tailed Student's $t$ test). **i** Bleeding times were repeated after bone marrow transplantation between Slc44a2(WT) and Slc44a2(KO) mice ($n = 10, 7, 9$, and 4 mice as shown in the chart ±S.D. and *$P < 0.01$ for WT–WT vs. KO–WT and *$P < 0.01$ for KO–KO vs. WT–KO in Tukey's range test). Bone marrow from Slc44a2(KO) donor mice prolongs the bleeding time of recipient mice. **j** Percent maximal blood flow in carotid artery after treatment with FeCl₃ was measured by ultrasound. **k** Quantitation of **j**. For WT–WT vs. KO–WT, the Fisher's exact test statistic is 0.02 and the result is significant at $P < 0.05$. For WT–KO vs. KO–KO, the Fisher's exact test statistic is 0.3 and the result is not significant at $P < 0.05$ Sample size includes: 10, 12, 7, 6, and 8 as shown in chart. **l** The bleeding time of Slc44a2(WT) and Slc44a2(KO) mice was measured after tail transection (WT and KO). ($n = 6, 9, 9$, and 8 mice as shown in the chart ±S.D. and *$P < 0.01$ for WT vs. KO). The bleeding time of Slc44a2(KO) mice after transfusion with platelets from Slc44a2(WT) or Slc44a2(KO) mice was measured after tail transection (WT to KO and also KO to KO). ($n = 10$ mice ±S.D. and *$P < 0.01$ for WT to KO vs. KO to KO compared in Tukey's range test). Source data are provided as a Source Data file.

**Slc44a2 regulates mitochondrial function**. Since our data indicate that Slc44a2(KO) platelets have abnormal mitochondrial ROS production, we next examined mitochondrial number and function in Slc44a2(KO) mice. We observed that platelets from Slc44a2(KO) mice contain more mitochondria by measuring mtDNA (Fig. 3a, b). However, platelets from Slc44a2(WT) and Slc44a2(KO) have similar numbers of alpha-granules and dense granules (Supplementary Fig. 1). We used a fractional immuno-blotting approach to show that Slc44a2 is expressed in mitochondria (Fig. 3c). Since Slc44a2 has functional homology with choline transporter proteins, we characterized choline transport into whole platelets and into mitochondria isolated from platelets. We incubated platelets or purified mitochondria with radiolabeled choline, or with radiolabeled choline in the presence of nonlabeled choline, washed the cells or mitochondria, and measured residual radioactivity. Slc44a2 mediates choline transport into mitochondria, but not into whole platelets (Fig. 3d, e). We performed mass spectrometry analysis of platelets from Slc44a2(KO) and Slc44a2(WT) mice to measure choline metabolites (Fig. 3f). Mice lacking Slc44a2 have decreased choline and one of its metabolites dimethylglycine (Fig. 3f). This suggests that Slc44a2 transports choline into mitochondria, and that choline is actively metabolized within mitochondria.

We next measured the effect of Slc44a2 upon mitochondrial function. A primary function of mitochondria is oxidative phosphorylation and ATP production, processes which consume

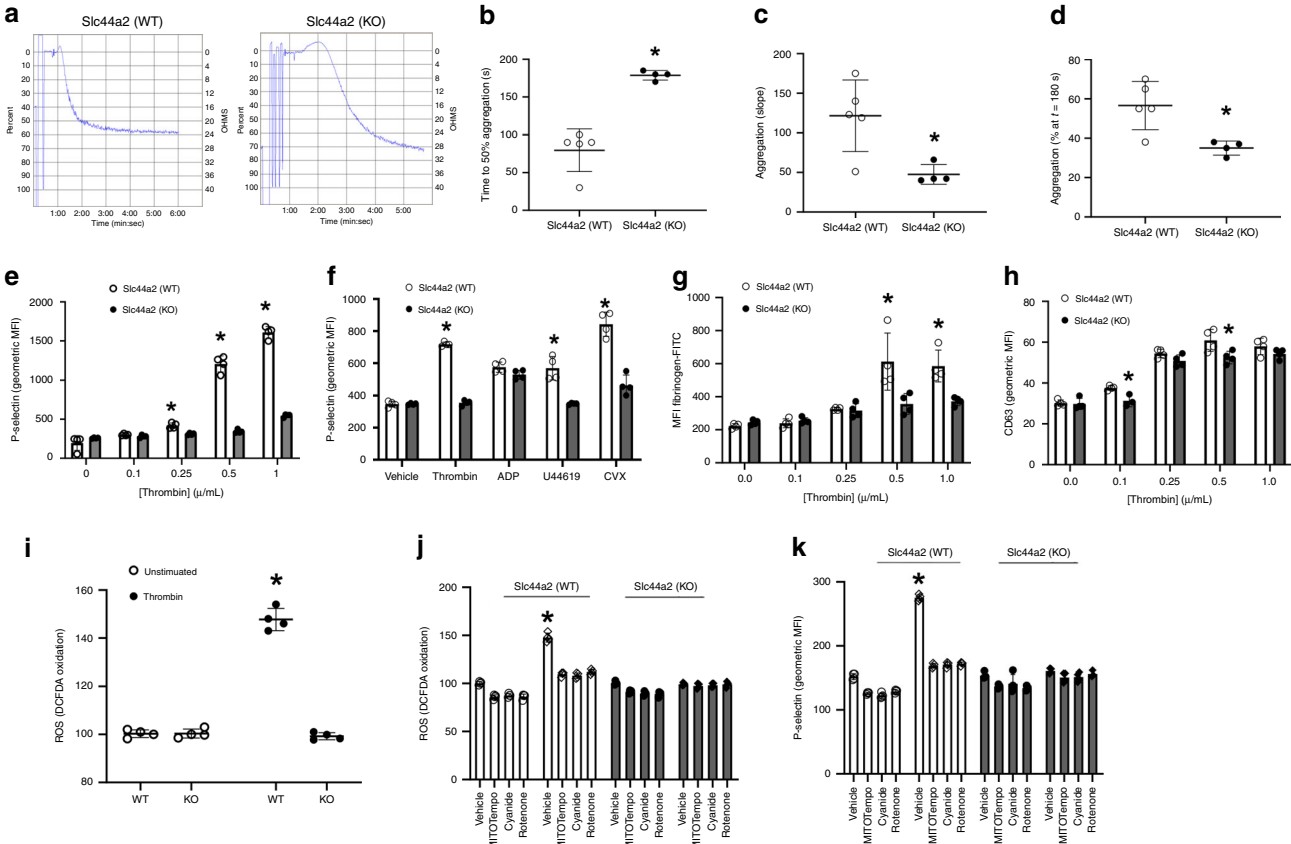

**Fig. 2 Slc44a2 regulates murine platelet activation ex vivo. a–d** Platelet aggregation after treatment with 0.5 μ/mL thrombin was measured by light transmission aggregometry. $n = 4$ biologically independent samples ±S.D. *$P < 0.05$ in a two-tailed Student's $t$ test. **e** Platelet externalization of P-selectin after thrombin stimulation was measured by flow cytometry. $n = 3$ biologically independent samples ±S.D. *$P < 0.05$ in a two-tailed Student's $t$ test. **f** Platelet externalization of P-selectin after treatment with various agonists (0.5 μ/mL thrombin, 5 μM ADP or U46619, or 0.5 ng/mL convulxin). $n = 3$ biologically independent samples ±S.D. *$P < 0.05$ in a two-tailed Student's $t$ test. **g** Platelet activation of GPIIbIIIA after 0.5 μ/mL thrombin treatment was measured by flow cytometry of FITC-fibrinogen binding. $n = 3$ biologically independent samples ±S.D. *$P < 0.05$ in a two-tailed Student's $t$ test. **h** Platelet externalization of CD63 after 0.5 μ/mL thrombin treatment was measured by flow cytometry. $n = 4$ biologically independent samples ±S.D. *$P < 0.05$ in a two-tailed Student's $t$ test. **i** Platelet ROS production during stimulation with 0.5 μ/mL thrombin was measured by flow cytometry with DCF-DA. **j** Platelet ROS production was measured after treatment with mitochondrial inhibitors followed by 0.5 μ/mL thrombin treatment. $n = 3$ biologically independent samples ±S.D. *$P < 0.05$ in a two-tailed Student's $t$ test. **k** Platelet externalization of P-selectin was measured after mitochondrial inhibitor treatment followed by 0.5 μ/mL thrombin treatment. $n = 3$ biologically independent samples ±S.D. *$P < 0.05$ in a two-tailed Student's $t$ test. For **j** and **k**, all compounds tested at 5 μM with thrombin at 0.5 μ/mL. Source data are provided as a Source Data file.

oxygen. Platelets from *Slc44a2(KO)* mice have a lower baseline of oxygen consumption rate than *Slc44a2(WT)* mice platelets, as measured by the Seahorse analyzer (Fig. 3g before oligomycin). Oxygen consumption rates following treatment with the ATP synthase inhibitor oligomycin are similar between *Slc44a2(KO)* and *Slc44a2(WT)* platelets, suggesting that they have similar rates of nonmitochondrial oxygen usage (Fig. 3g, after oligomycin treatment). The maximum rate of ATP production is greater in *Slc44a2(WT)* platelets than in *Slc44a2(KO)* platelets, following treatment with the membrane uncoupler FCCP (carbonyl-cyanide-4-(trifluoromethoxy) phenylhydrazone) (Fig. 3g after FCCP). When rotenone and antimycin A are added to inhibit electron transport, the oxygen consumption rate falls to equally low levels for both *Slc44a2(WT)* and *Slc44a2(KO)* platelets (Fig. 3g after Rot/AA). Thus, resting platelets lacking Slc44a2 consume less oxygen, and display a decreased maximum oxygen consumption after FCCP treatment (Fig. 3g). Finally, we observed that the addition of choline produces a moderate increase in oxygen consumption in *Slc44a2(WT)* platelets, but not in *Slc44a2 (KO)* platelets (Fig. 3h). These data suggest that choline plays a major role in mitochondrial respiration.

**Slc44a2 increases mitochondrial production of ATP.** Platelets secrete ADP after stimulation by multiple extracellular agonists such as thrombin[19]. After its release, ADP then amplifies platelet activation in an autocrine and paracrine manner through the platelet P2Y12 receptor[20]. Since we observed decreased mitochondrial function in *Slc44a2(KO)* platelets, we next compared ATP and ADP levels in isolated platelets. Platelets from *Slc44a2 (KO)* mice contain less ADP and less ATP than platelets from *Slc44a2(WT)* mice (Fig. 4a, b). The ratio of ADP/ATP in Slc44a2 (KO) platelets is less than in *Slc44a2(WT)* platelets (Fig. 4c). We found that platelets from *Slc44a2(KO)* mice release less ATP after thrombin stimulation than *Slc44a2(WT)* platelets (Fig. 4d). We tested the ability of choline to influence mitochondrial ROS and ATP production: mitochondria from *Slc44a2(WT)* platelets produce additional ROS and ATP in response to choline, but mitochondria from *Slc44a2(KO)* platelets fail to make additional ROS or additional ATP in response to choline (Fig. 4e, f).

Decreased ATP production and release could explain why *Slc44a2(KO)* platelet activation is diminished, since ATP and ADP released from platelets boosts platelet activation. If this is true, then adding exogenous ADP should rescue the defective

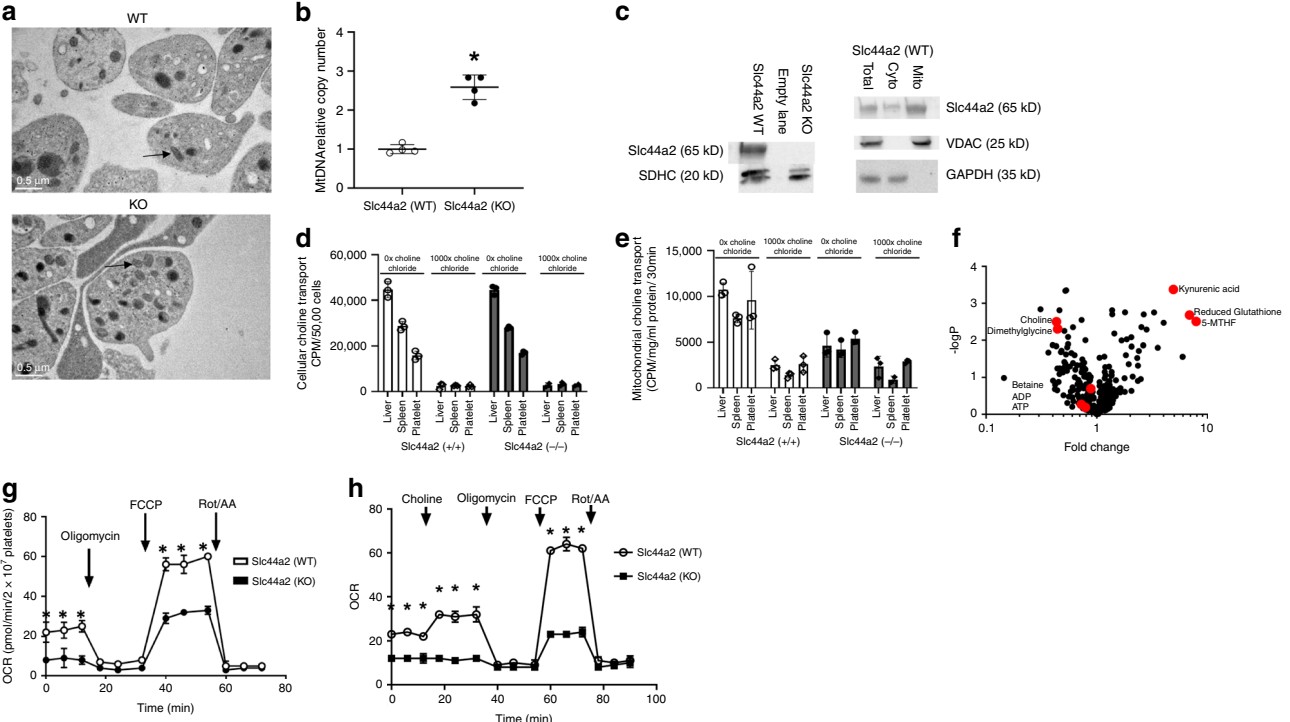

**Fig. 3 Slc44a2 regulates mitochondrial number and function. a** Representative transmission electron microscopy images of platelets from *Slc44a2(WT)* and *Slc44a2(KO)* mice. Arrow indicates an individual mitochondrion. **b** Platelet mitochondria DNA copy number measured by qPCR. *n* = 4. *P < 0.05 in a two-tailed Student's *t* test (**c**) Slc44a2 is present in lysate of mitochondria purified from platelets (left immunoblot) and Slc44a2 is enriched in the mitochondrial fraction of platelets but not in the cytosolic fraction of platelets (right immunoblot). **d** Slc44a2 does not regulate choline transport into whole cells, as measured by cellular uptake of radiolabeled choline competed with nonlabeled choline. (*n* = 3 biologically independent samples ±S.D. and *P < 0.05 for WT vs. KO). **e** Slc44a2 regulates transport of choline into isolated mitochondria, as measured by mitochondrial uptake of radiolabeled choline competed with nonlabeled choline. (*n* = 3 biologically independent samples ±S.D. and *P < 0.05 for WT vs. KO). **f** Metabolite profiles in platelets from *Slc44a2(KO)* mice relative to *Slc44a2(WT)* mice as measured by mass spectroscopy. (*n* = 2–3). **g** Mitochondrial oxygen consumption rate (OCR) of platelets from *Slc44a2(KO)* and *Slc44a2(WT)* mice: basal and uncoupled OCR are decreased in *Slc44a2(KO)* platelets (*n* = 4 biologically independent samples ±S.D.; *P < 0.05 WT vs. KO in a two-tailed Student's *t* test). **h** Mitochondrial OCR during treatment with choline (*n* = 4 biologically independent samples ±S.D.; *P < 0.05 WT vs. KO in a two-tailed Student's *t* test). For stress testing, Antimycin A = 1.0 μM, FCCP = 1.0 μM and Rotenone = 0.5 μM. Choline was added at 20 μM for all experiments. (For all panels, *P < 0.05 WT vs. KO in a two-tailed Student's *t* test). Source data are provided as a Source Data file.

activation of platelets lacking Slc44a2. To test this hypothesis, we added exogenous nonhydrolyzable ADP to platelets. As before, thrombin failed to activate platelets from *Slc44a2(KO)* mice, but 2-me-ADP restores activation of *Slc44a2(KO)* platelets in response to thrombin (Fig. 4g). We also tested a nonhydrolyzable ATP analog, and this agonist slightly increases platelet activation as expected, and there is no difference between WT and KO platelets (Supplementary Fig. 2). Taken together, our data suggest that Slc44a2 controls intracellular levels of ATP and ADP, and abnormally low platelet levels of ADP and ATP contribute to the impaired activation of platelets from *Slc44a2(KO)* mice.

## Discussion

The major findings of our study are that choline is a substrate for mitochondria, Slc44a2 regulates choline transport into mitochondria, and choline plays a key role in production of ATP necessary for maximal platelet activation (Fig. 5).

We discovered that mitochondria in *Slc44a2(KO)* platelets are dysfunctional: their oxygen consumption rate at rest is low, and their maximal respiration rate when uncoupled is also low, compared with wild-type controls. Mitochondrial dysfunction leads to decrease ATP content in platelets, decreased ATP release from platelets, and decreased extracellular ADP. Decreased ADP in turn leads to less paracrine and autocrine

platelet activation. One potential explanation for this defect in platelet activation is that Slc44a2 mediates choline transport into mitochondria, where it is metabolized into substrates for oxidative metabolism which generate ATP. Another possibility is that Slc44a2 mediates choline transport into mitochondria where it is metabolized into structural compounds such as phosphatidylcholine phospholipids that stabilize mitochondrial membranes which are necessary for oxidative phosphorylation[16,17,21]. Slc44a2 may also regulate platelet activation by controlling mitochondrial production of ROS, which are important for mediating platelet activation[18,22–30].

GWAS of human subjects have shown that genetic variants in the *SLC44A2* locus are linked to abnormalities in risk of VTE[3,4]. Our data support these findings by demonstrating that Slc44a2 regulates platelet activation and thrombosis in mice (Figs. 1 and 2). The human GWAS data also show that the genetic variants in the *SLC44A2* locus are not associated with known hemostatic biomarkers[3,4].

Our study of Slc44a2 is supported by prior work. Others have also shown that Slc44a2 increases thrombosis in various murine models[11,13]. Searching for mechanistic pathways through which Slc44a2 affects thrombosis, one study excluded an effect of Slc44a2 upon plasma proteins including coagulation factors, and proposed that Slc44a2 affects thrombosis through cellular pathways[12]. We confirm this prediction and extend this prior work,

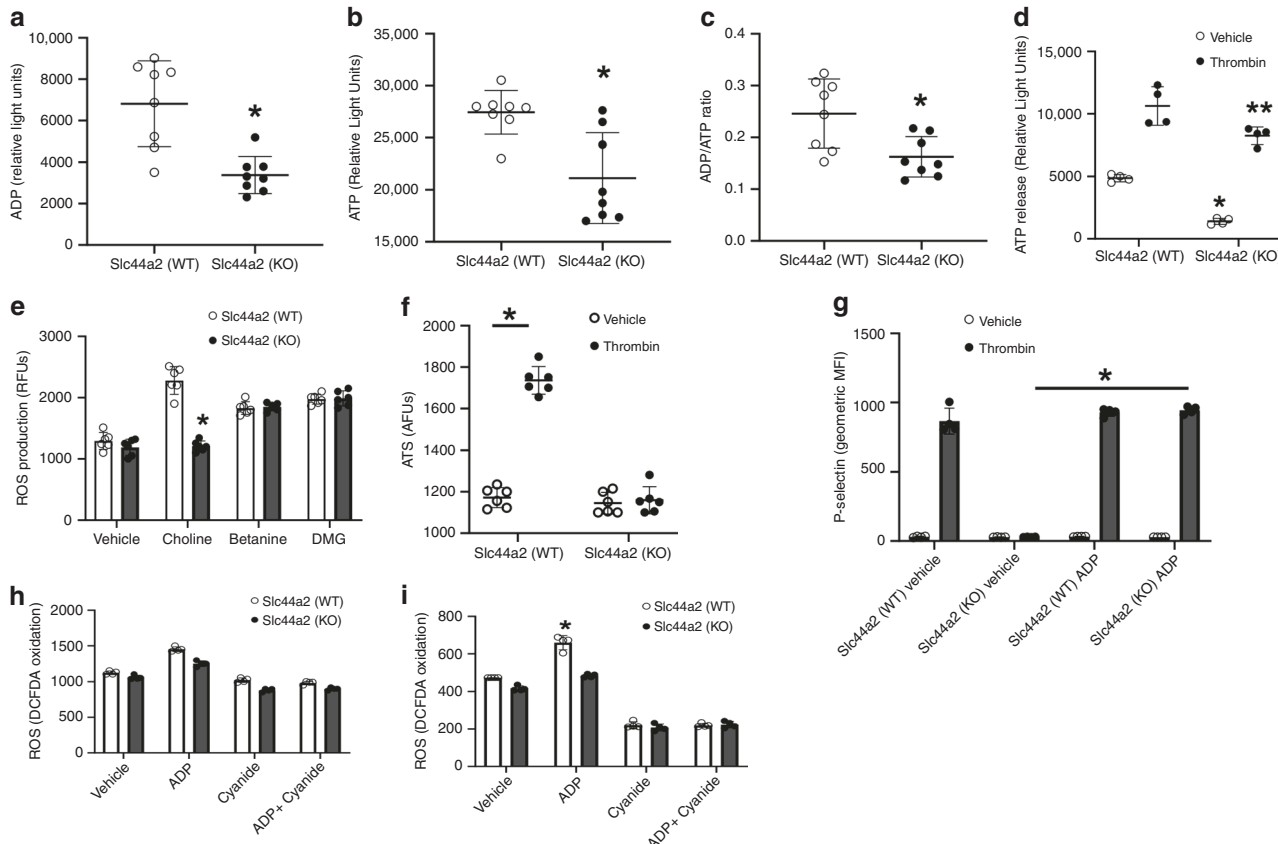

**Fig. 4 Slc44a2 and choline increase mitochondrial production of ATP and platelet ATP/ADP levels. a** Platelets from *Slc44a2(KO)* mice contain less ADP than platelets from *Slc44a2(WT)* mice. $n = 7$ biologically independent samples ±S.D. *$P < 0.05$ in a two-tailed Student's *t* test. **b** Platelets from *Slc44a2(KO)* mice contain less ATP than platelets from *Slc44a2(WT)* mice. $n = 7$ biologically independent samples ±S.D. *$P < 0.05$ in a two-tailed Student's *t* test. **c** Platelets from *Slc44a2(KO)* mice have an altered ADP/ATP ratio compared to platelets from *Slc44a2(WT)* mice. $n = 7$ biologically independent samples ± S.D. *$P < 0.05$ in a two-tailed Student's *t* test. **d** Platelets from *Slc44a2(KO)* mice release less ATP in response to 0.5 μ/mL thrombin. $n = 4$ biologically independent samples ±S.D. *$P < 0.05$ in a two-tailed Student's *t* test. **e** Choline increases ROS production in a manner that depends upon Slc44a2 in purified mitochondria. $n = 3$ biologically independent samples ±S.D. *$P < 0.05$ in a two-tailed Student's *t* test. **f** Choline increases ATP production in a manner that depends upon Slc44a2 in isolated mitochondria. Choline was added at 20 μM. $n = 6$ biologically independent samples ±S.D. *$P < 0.05$ in a two-tailed Student's *t* test. **g** Exogenous ADP rescues platelet activation defect in *Slc44a2(KO)* platelets. Platelets were treated with 0.5 μ/mL thrombin and 5 μM 2-MeSADP. $n = 3$ biologically independent samples ±S.D. *$P < 0.05$ in a two-tailed Student's *t* test. **h** ADP increases platelet ROS production. **i** ADP increases mitochondrial ROS production. $n = 3$ biologically independent samples ±S.D. *$P < 0.05$ in a two-tailed Student's *t* test. Source data are provided as a Source Data file.

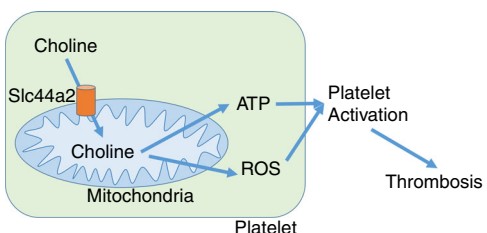

**Fig. 5 Proposed model for choline transport and Slc44a2 regulation of platelet activation.** We propose that Slc44a2 transports choline into mitochondria, where it is metabolized and regulates the production of ATP and release of ROS. ATP is released from platelets and hydrolyzed to ADP which acts upon platelet purinergic receptors and drives further platelet activation.

now demonstrating that Slc44a2 promotes thrombosis by regulating platelet metabolism.

Our study has several limitations. We show that Slc44a2 directly affects platelet function, since platelets isolated from

*Slc44a2(KO)* mice have activation defects (Fig. 2) and since platelet transfusion partially restores bleeding time in *Slc44a2(KO)* mice (Fig. 1). However, other cells lacking Slc44a2 may also contribute to the hemostatic and thrombotic abnormalities in *Slc44a2(KO)* mice. For example, knockout mice have lower lymphocyte counts compared to wild-type mice (Supplementary Table 1), and lymphocytes can regulate thrombosis in vivo[31–33]. Changes in the number of lymphocytes can alter the kinetics and composition of DVT formation as a "red thrombus" with a substantial portion of erythrocytes in the thrombus. As another example, even though *Slc44a2(KO)* mice have neutrophil counts similar to WT mice (Supplementary Table 1), others have shown that neutrophils can influence thrombosis[34–38]. Changes in the number of neutrophils can alter arterial thrombus, which appears as a platelet-rich "white thrombus." It is possible that Slc44a2 expressed in leukocytes can partially influence thrombosis in the venous and arterial beds. Our data show that *Slc44a2(KO)* mice have decreased hemostasis and thrombosis, and we show that *Slc44a2(KO)* platelets have decreased activation, and transfusions of platelets from *Slc44a2(KO)* mice fail to improve the bleeding

time. These data suggest that Slc44a2 in platelets is a major cause of the hemostasis defect in *Slc44a2(KO)* mice.

In conclusion, our data suggest that Slc44a2 is a thrombosis regulator which does not lie within a well-studied thrombotic pathway, and Slc44a2 may be an important therapeutic target.

## Methods

**Mice.** *Slc44a2(KO)* mice were obtained from Dr. Thomas Carey (University of Michigan) and a colony was established at the University of Rochester[8]. These mice are global *Slc44a2* null mice that lack exons 3–10 of *Slc44a2*[8]. All mice were treated in accordance with established protocols at the University of Rochester, which are approved by the University Committee on Animal Research (UCAR). These approved protocols and all studies with mice complied with all ethical regulations for animal testing and research at the University of Rochester. This study received ethical approval by the UCAR of the University of Rochester. All experiments used male mice aged 8–12 weeks, except for mesenteric thrombosis where mice 6 weeks of age were chosen to facilitate visualization of the mesenteric arteries or venules. Every effort was taken to minimize pain and discomfort in the mice used for breeding or experiments.

**Antibodies and reagents.** Primary antibodies against Slc44a2 were purchased from Aviva Systems Biology (ARP44009_P050) and used at a dilution of 1:500. Primary antibody against GAPDH was purchased from Abcam (ab 8245) and used at a dilution of 1:2000. HRP-conjugated secondary antibody against rabbit primary antibody were purchased from GE (NA931) and used at a dilution of 1:5000. Anti-CD62-PE and Anti-CD63-PE antibodies for flow cytometry were purchased from BioLegend (Catalog # 304905 and #143903). FITC-Fibrinogen was purchased from Thermo Fisher (Catalog # RB-1924-R2). DCF-DA was purchased from Invitrogen. All ROS enzyme inhibitors were purchased directly from Sigma-Aldrich.

**Western blotting.** Western blots were performed as previously described[39]. In brief, endothelial cells or platelets were lysed with Laemmli sample buffer (Bio-Rad), boiled for 5 min at 95 °C, resolved on 4–20% Mini-PROTEAN TGX Precast gels (Bio-Rad), and transferred using a wet electrophoretic transfer unit (Bio-Rad) onto nitrocellulose membranes. After 1-h blocking with 5% nonfat milk in PBS containing 0.05% Tween 20 at room temperature, the membranes were hybridized with primary antibodies followed by HRP-conjugated secondary antibodies and enhanced chemiluminescence detection using a Bio-Rad Chemidoc Imager and ImageLab software (Bio-Rad) (see Supplementary Fig. 3).

**Quantitative real-time PCR.** Murine tissue was harvested immediately after sacrifice and snap frozen on dry ice. Total RNA was isolated using an RNEasy kit following the manufacture's protocol (Qiagen). The A260/A280 ratio of all samples was between 1.9 and 2.1 as measured by spectrophotometry (NanoDrop; Thermo Scientific). cDNA was synthesized using an iScript™ cDNA Synthesis kit (Bio-Rad). Quantitative real-time PCR was performed by PrimePCR Sybr Green gene expression assay (Bio-Rad) for 40 cycles on a CFX Connect thermal cycler (Bio-Rad). The qPCR primers were form a kit (Bio-Rad qMmuCID0026393 and qMmuCED0027497). Quantification was performed in triplicate for each sample, and tissues were harvested from at least $n = 3$ mice. Expression results were calculated by $\Delta\Delta CT$ method and were normalized to the reference gene *Gapdh*.

**Platelet isolation and activation.** Murine blood was obtained by retro-orbital bleeding of anesthetized animals into heparinized murine Tyrode's buffer (134 mM NaCl, 2.9 mM KCl, 12 mM NaHCO$_3$, 0.34 mM Na$_2$HPO$_4$, 20 mM HEPES, pH 7.0, 5 mM glucose, and 0.35% bovine serum albumin) in Eppendorf tubes. The blood was then centrifuged to yield platelet-rich plasma which was then washed in new tubes containing Tyrode's buffer with 1% PGE$_2$ to prevent platelet activation. Platelets were then pelleted by 5 min $600 \times g$ centrifugation at room temperature and the supernatant was discarded. The pelleted platelets were gently resuspended in Tyrode's buffer and kept at room temperature for further experiments within 2 h. For platelet activation, diluted platelet suspension was divided into 100 μL aliquots, treated with indicated drugs where appropriate (10 μM MitoTEMPO, 1 mM NAC, 100 μM sodium cyanide, or 10 μM rotenone), and then stimulated with PBS or indicated concentrations of thrombin for 15 min followed by staining with 1 μL CD62P-PE/CD63-PE (BioLegend) or 1 μL Fibrinogen-Oregon Green (Thermo Fisher) for 30 min, and immediately fixed with 100 μL 2% formalin. For DCFDA experiments, platelets were loaded with 10 μM DCFDA at 37 °C for 1 h, and then treated/stimulated as above. The fluorescence intensity was measured on an Accuri C6 Flow Cytometer (BD Biosciences). The data were analyzed with FlowJo software (Tree Star Inc.). ATP release was measured in poststimulation supernatant using an ATP Bioluminescent Assay Kit (Sigma-Aldrich). ADP/ATP contents of whole platelets was determined using an ADP/ATP Ratio kit, according to manufacturer instructions (Abcam). Aggregometry was performed using flow cytometry as described[40]. Mitochondrial DNA copy number was measured in 50,000 platelets by qPCR using a commercial kit (Abcam), and normalized to WT copy number.

**Mouse tail bleeding assay.** Mouse tail bleeding time was measured as described[41]. After IP anesthetization with ketamine and xylazine (80/12 mg/kg), the distal 3 mm of the tails of the mice were amputated and immersed immediately in 37 °C saline, and the time to visual cessation of bleeding for 30 s or continuous bleeding to 20 min maximal duration, whichever occurs first, was recorded.

**Mouse mesenteric thrombosis model and carotid thrombosis model.** Thrombosis was measured as previously described[42,43]. For the mesenteric thrombosis model, mice were anesthetized and platelets labeled with DyLight488-antibody to GPIb beta. A target area containing mesenteric arterioles (120–150 μm in diameter) was externalized for imaging. The arteriole flow was recorded for 3 min at resting condition. Then 1 mm$^2$ of Whatman paper saturated with 7.5% FeCl$_3$ solution was applied to the arteriole for 3 min and the arteriole flow was continuously recorded for a total of 20 min The time to form a small thrombus (50-pixel diameter) and to full vessel occlusion were recorded. Recording was terminated at the end of 20 min if no occlusion were observed. For the carotid thrombosis model, mice were sedated with 2.5% isoflurane and maintained anesthetized with 2% isoflurane. The common carotid arteries were exposed for a baseline flow recording using an MA1PRB Perivascular Flowprobe and a TS420 Flowmeter (Transonic Systems). Then $1 \times 2$ mm Whatman paper soaked with 1.5 μL 7.5% FeCl$_3$ solution was applied to the ventral surface of the carotid upstream of the flowprobe for 3 min Flow measurement was resumed for a total of 20 min after FeCl$_3$ wash off. We define occlusion as the absence of blood flow (0 mL/min) for 3 min

**Mouse deep vein thrombosis model.** For the DVT model, mice were anesthetized, a midline incision was made. A ligature was secured around the IVC between the renal veins and the bifurcation of the iliac veins, and a 28-gauge blunt needle was placed between the ligature and the IVC. The suture was tied off, temporarily causing complete occlusion of the IVC, and the needle was removed from under the suture, restoring partial luminal flow with a severe stenosis of the IVC. The wound closed with sutures. Six hours following ligation, the mice were anesthetized and the IVC was isolated including any thrombus which had formed. Total mass was recorded, and the isolated IVCs deposited into 4% PFA for 24–48 h, and then the thrombus was excised and the thrombus mass was recorded.

**Bone marrow transplantation.** *Slc44a2(WT)* and *Slc44a2(KO)* donor mice were euthanized, and femurs were isolated under sterile conditions. Bone marrow was harvested, and then aspirated repeatedly to create single cell suspensions. Cell counts were manually performed and 10$^7$ cells were injected into each recipient mouse intravenously via the retro-orbital plexus on the same day as lethal irradiation of recipients. Eight-week-old mice were used as recipients and were lethally irradiated with an X-ray RS 2000 (Rad-sources) irradiator delivering a split dose of 1100 rad, delivered as 550 rad in the morning and then 550 rad in the afternoon the day of transplantation. After marrow transplantation, mice were provided with water supplemented with sulfatrim for 2 weeks, and allowed to reconstitute for 6 weeks prior to tail bleeding assay or blood collection.

**Platelet transfusion.** Platelets were isolated as above. Washed platelets were transfused into recipient mice through a retro-orbital injection[44].

**Mitochondrial isolation and ATP production.** Isolation of mitochondria was performed using a commercially available kit (Thermo Fisher #89801). Briefly, platelets are isolated as described above, lysed and then a series of sucrose gradient centrifugation steps performed to yield a pure layer of mitochondria, which was then suspended for further experiments. For ATP production, mitochondria were suspended in mitochondrial respiration buffer KCl (120 mM), sucrose (25 mM), HEPES (10 mM), EGTA (1 mM), KH$_2$PO$_4$ (1 mM), MgCl$_2$ (5 mM), glutamate (15 mM), and malate (7.5 mM) pH 7.3, supplemented with vehicle or choline, and ATP quantified as above.

**Choline transport.** The rate of choline uptake was determined by measuring 3H-choline chloride (Perkin Elmer) uptake over time. Platelets or mitochondria were added into 1.5 mL tubes. One hour prior to uptake, media were removed and cells were washed with PBS before being incubated in Tyrode's buffer until use. Immediately prior to uptake, cells were washed again followed by the addition of Tyrode's buffer containing 1 μCi/mL of 3H-choline and were incubated at 37 °C for desired time point (1–30 min). Following incubation, cells were washed twice with Tyrode's buffer, lysed in 150 μL of 0.1 M NaOH and an aliquot was used to determine radioactivity by liquid scintillation counting. Total cellular protein was determined using a BCA protein assay (Thermo Fisher Scientific) according to the manufacturer's instructions. Choline uptake was expressed as counts per minute per mg protein (for mitochondria) or per 50,000 cells (for platelets).

**Platelet mitochondrial function measurements.** A 96-well format Seahorse extracellular flux analyzer (Seahorse Bioscience, MA, USA) was used to measure bioenergetics[45,46]. Platelets were diluted to a concentration of $2 \times 10^7$ in XF DMEM assay buffer (DMEM with 1 mM pyruvate, 5.5 mM D-glucose, 4 mM L-glutamine, pH 7.4) and were seeded onto Cell-Tak coated XF96 microplates and

mitochondrial stress test was performed as described[45,46]. For some Seahorse experiments, choline was injected at 10 µM prior to the stress test.

**Mass spectrometry**. We used LC-MS/MS based methods to profile 147 analytes including amino acids, organic acids, bile acids, indoles, nucleotides, and sugars. We have previously used these platforms to characterize biochemical pathways implicated in metabolism[47,48]. Briefly, samples were deproteinized using extraction solvent containing stable isotope labeled internal standards. Samples were vortexed and centrifuged, and aliquots were transferred to 2 mL autosampler vials with glass inserts for LC-MS analysis. In positive mode, normal phase hydrophilic interaction chromatography (HILIC) using a $2.1 \times 150$ mm 3 µm Atlantis column (Waters) was coupled to a 4000 QTrap triple quadrupole mass spectrometer (Applied Biosystems/Sciex) equipped with an electrospray ionization source for targeted detection of 78 metabolites using a dynamic multiple reaction monitoring (dMRM) mechanism. In negative mode, HILIC chromatography using a $2.1 \times 100$ mm 3.5 µm Xbridge Amide column (Waters) was coupled to an Agilent 6490 triple quadrupole mass spectrometer equipped with an electrospray ionization source for targeted detection of 69 metabolites using dMRM. Metabolite peak areas were integrated using Sciex MultiQuant software (positive mode) or Agilent Masshunter Quantitative software (negative mode). All metabolite peaks were manually reviewed for peak quality in a blinded manner. (See Supplementary Tables 2 and 3 for metabolites and metabolomic data.)

**Human platelet collection**. Human blood collection was performed as previously described using protocols approved by the Institutional Review Board at the University of Rochester Medical Center (IRB Protocol RSRB00028659)[49]. Normal healthy blood donors were recruited. Subjects were excluded if they had used aspirin or any nonsteroidal anti-inflammatory agent within 10 days before the blood draw. Blood was collected by venipuncture into sodium citrate anticoagulant tubes. Whole blood was centrifuged at $180 \times g$ for 15 min to isolate the top layer of platelet-rich plasma, which was diluted 1:20 in room temperature Tyrode's buffer (134 mM NaCl, 2.9 mM KCl, 12 mM NaHCO3, 0.34 mM Na2HPO4, 20 mM HEPES, pH 7.0, 5 mM glucose, and 0.35% bovine serum albumin) and dispensed in 100-µL volumes for treatment with various drugs. Western blotting was performed as previously described[49].

**Statistics**. Data with a normal distribution were analyzed by two-tailed Student's $t$ test for comparison of two groups, and by ANOVA to compare means of three or more groups. Statistical significance was defined as $P < 0.05$. To compare the tail bleeding time of two groups (bleeding time for *Slc44a2(WT)* and *Slc44a2(KO)* mice in Fig. 1e), a dichotomous measure of cessation of bleeding within 20 min (yes or no) was used and compared in a $2 \times 2$ contingency table. Because of small sample sizes within some of the cells, the Fisher's Exact Probability test was used to compare the two groups. To compare the thrombosis time of two groups (time for cessation of flow in the carotid artery for WT to KO and KO to KO mice in Fig. 1k), a dichotomous measure of thrombosis within 20 min (yes or no) was used and compared in a $2 \times 2$ contingency table. Because of small sample sizes within some of the cells, the Fisher's Exact Probability test was used to compare the two groups. Tukey's test was used for multiple comparisons (such as differences in bleeding time between transfused hosts in Fig. 1l).

**Study approval**. All in vivo procedures and usage of mice were approved by the Division of Laboratory Animal Medicine at the University of Rochester Medical Center.

**Reporting summary**. Further information on research design is available in the Nature Research Reporting Summary linked to this article.

## Data availability

The datasets generated during and analyzed during the current study are available from the corresponding author on reasonable request. The dataset of the metabolomics experiment is available in the Supplementary Material. The source data underlying Figs. 1a–e, g–l, 2b–k, 3c–e, g, h, and 4a–i are provided as a Source Data file. Source Data are provided with this paper.

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

## Acknowledgements

This work was supported by NIH NHLBI R01 HL134894 (C.J.L. and N.L.S.), R61 HL141791 (C.J.L. and N.L.S.), R01 HL124018 (C.N.M.), K08 HL128856 and NIH HL12020 (S.J.C.), R01 HL139553 (N.L.S.), R01 CA194536 (T.E.C.), U01 DE025184 (T.E.C.), R01 HL071158 (P.S.B.), a Miguel Servet contract from the ISCIII Spanish Health Institute CP17/00142 and from the European Social Fund (M.S-L.), a gift from Richard T. Aab and the Aab Cardiovascular Research Foundation, and the Paul N. Yu Professorship (C.J.L.). Electron Microscopy support was provided by the Electron Microscopy Core Facility at University of Rochester.

## Author contributions

J.A.B. designed experiments, conducted experiments, acquired data, analyzed data, and wrote the manuscript. S.K.T. conducted experiments and trained lab workers in animal procedures. M.A.M. conducted experiments, acquired data, and analyzed data. R.A.B. conducted experiments and trained lab workers in platelet analyses. S.J.C. designed experiments, conducted experiments, acquired data, and analyzed data. T.E.C. provided reagents and reviewed the manuscript. S.G.L. and R.A.B. and S.L.S assisted with the platelet aggregation studies. P.S.B. and C.O.S. and J.A.B. designed and assisted in Sea-horse analyzer studies. R.E.G. and X.S. designed and performed metabolomics studies. R.M.B. conducted experiments. N.L.S. and M.S.-L. and P.S.d.V. and J.E.H. reviewed the manuscript. C.N.M. designed experiments, analyzed data, and wrote the manuscript. C.J.L. designed experiments, analyzed data, and wrote the manuscript.

## Competing interests

The authors declare no competing interests.
