## [Peer Review File · Nature Communications]

Reviewers' Comments:

Reviewer #1:

Remarks to the Author:

General comments:

This is a generally well-written manuscript in which the authors, through a set of cleverly designed experiments, demonstrate that the choline transporter Slc44a2 controls platelet activation through a mechanism involving mitochondrial production of ATP. This study unravels a new regulator of platelet activation that might be eventually used for the modulation of diseases involving an abnormal platelet aggregation. To demonstrate their hypothesis, Bennet et al. take advantage of the use of a Slc44a2 knockout mice, that has been demonstrated to have impaired platelet activation. We highlight the number of different techniques used in the study, from molecular biology, to sub-cellular fractioning or mass spectrometry. The order of experiments flows reasonably well, allowing the reader to follow the paper easily. Therefore, many of the possible questions that arise from a set of results are readily solved in the next figure. However, the presentation of the data is poor and somehow confusing, not matching its quality. This is an issue that should be improved in the next versions of the manuscript.

This work provides a well-designed approach for the characterization of a molecular target of platelet function beyond the thrombotic pathway, as well as, an endogenous metabolite (ADP) that can modulate platelet activation when it is added exogenously. All in all, we do not think that there is any major concern that precludes the publication of this article. However, several minor concerns should be adequately addressed before its publication.

Minor concerns:

1. Presentation of data should be improved in a global way. Resolution of figures is low, colors are sometimes difficult to distinguish, and doses of some molecules are missed. We encourage the authors to improve the quality of figures, facilitating readers to follow them more easily. Also, captions should contain more information. In particular, the next points should be addressed:
 - For all the drugs/inhibitors added, the dose should be detailed either in the graph or in the caption. This applies at least for figures 2a, 2c, 2f-h, 3-h-j, 4d, 4f-g.
 - Captions information should be enough to minimally understand the experiment. Please, avoid descriptions like "Platelet GPIIbIIIa activation" (2d) and similar.
 - Figure 1c→ The WB quality is very low. It seems that you have the spots for the first and third condition and a blank hole in the middle. Please, repeat the WB or show a better image. Also, explain why HEK293 are used as control in the text or caption.
 - Figure 1d→ There is redundant information (legend and X-axis showing the same information).
 - Figure 1e→ Explain in the text (last paragraph of page2) that you are transplanting bone marrow cells between mice.
 - Figure 1f→ Upper panel is impossible to read. Make it bigger.
 - Figure 1g→ Time unit is shown as "sec" when in panel d is shown as seconds. Please unify, or even better, show international system notation (s).
 - Figure 1f→ X-axis labeling is not aligned with the bars. Rotate it vertically.
 - Figure 1a→ Impossible to see anything.
 - Figure 2g-h→ Some bars are thicker than others. Very confusing to follow in general. Please, use colors to help with the identification of which bar corresponds to which treatment.
 - Figure 3e-f→ If the magnitude measured is supposed to be the same (choline transport), why the Y-axis units are different? Please, clarify this. In order to distinguish between the 2 panels, indicate that the second one is isolated mitochondria (even when it is expressed in the caption).
 - Figure 3g→ Use different colors for dots that you consider of interest for the project instead of using arrows. Show the legend with the color of every metabolite. Make it bigger.
 - Figure 4g→ Use colors for the bars. The second and third one can be barely distinguished. I do not know if that grouping is the best way to make your data more impactful. I encourage you to figure out another way to make that graph with 3 variables (WT/KO, ADP and Thrombin) easier to follow.
2. A scheme with the proposed working model should be included as new figure 5. Since the data

presented in this manuscript supports a complex model of platelet activation through a tightly regulated mitochondrial function, the inclusion of a figure with the model proposed for Slc44a2 will clearly help readers to have a global idea without paying much attention to the results.

3. In figure 1a, we missed the expression of Slc44a2 in bone marrow cells. Especially, because bone marrow cell transplants are carried out in other experiments. Please include it. We would also like to have an explanation of why Slc44a2 expression is that high in heart. Could it be because β -actin is lower in heart? Do you think that using β -actin as housekeeping was the best choice for this experiment?

4. In figure 2c, different agonists are used to stimulate platelets, observing that ADP has an impact in KO mice. Have you included any reference that explains why ADP is pivotal for platelet activation? Otherwise, it seems that the fact of including ADP was a random choice.

In the line 10 of page 8, it is stated that "adding exogenous ADP should rescue the platelet activation decrease". In that particular form it seems that the idea of adding ADP arises as a consequence of the previous results (which is true in part). But also, it is perfectly known that ADP is a key player in platelet activation. Please include this fact at any point of the text. We also request the authors to show the effect of an analogous non-hydrolysable molecule of ATP in platelet activation, since the rationale to add ADP is the same to add ATP.

5. In the second line of page 6, please replace "spectrometric" by "spectrometry".

6. In the first paragraph of page 6, it is stated that "Mice lacking Slc44a2 have decreased choline and its metabolites such as betaine and dimethylglycine". Observing the figure 3g, it seems that betaine change is not statistically significant. With our interpretation of the Y-axis unit, when $-\log P < 1$, the pvalue is greater than 0.1, so betaine is not (by far) statistically significant. Is this correct? If so, please modify the text. Actually, I would say that betaine is not changed at all (fold change of 0.95 cannot be considered relevant).

7. In the second paragraph of page 7, FCCP is not introduced. It should be specified why you are using this compound for your assay.

8. In general, the results are powerful, but their discussion is vague and for some points is even incomplete. For instance, is it possible to find a connection between choline (and phosphatidylcholine phospholipids) metabolism and cardiolipins? Cardiolipins are very important lipids in the inner mitochondrial membrane and they have been demonstrated as key molecules for the oxidative phosphorylation. Also, patients with anti-cardiolipin antibodies can present thrombotic events.

Reviewer #2:

Remarks to the Author:

This is an interesting and in general well-preformed study. Overall, the data is of high quality and supportive of the overall conclusions. I have a few suggestions the authors might wish to consider. The authors state that 'Taken together, our data suggest that Slc44a2 control mitochondrial production of ROS during platelet activation'. This conclusion is based I believe on the data Figure 2g where the KO platelets do not produce an increase in ROS levels following thrombin stimulation. However, I would caution the authors from this strong of a conclusion since the KO cells do not respond at all to thrombin (i.e. no P-selectin, fibrinogen, etc.). As such, it is unclear whether there is a direct effect of Slc44a2 on ROS levels or a more general effect on thrombin-mediated activation. Looking at mitochondrial ROS levels in response to ADP might resolve this issue since Slc44a2 KO platelets respond to this stimulus. Secondly, while the overall data quality is high, I have two issues the authors can hopefully resolve. First, the change in ATP and ADP in Fig 3g is not significant while in Figure 4 it is. Is there a reasonable explanation for this? Secondly, the response to choline in Fig 3j looks quite modest. Is this significant/reproducible? Finally, the data in Figure 3g is of interest and if possible should be shown in a more usable form as a Supplemental Table as well as in its current volcano plot.

Reviewer #3:

Remarks to the Author:

In this study entitled, "The choline transporter Slc44a2 controls platelet activation and thrombosis by regulating mitochondrial function", Bennett and colleagues propose the choline transporter Slc44a2 as a novel therapeutic target based on the observation that Slc44a2-deficient mice display a prolonged time to in vivo thrombus formation. Furthermore, the authors link mitochondrial metabolism to platelet activation.

In general, the data shown in this manuscript is interesting but at the present stage considered preliminary. Further, the conclusions are not always fully supported by the presented data. I have the following major concerns:

1. An Abstract summarizing the main findings of the manuscript is missing.
2. A detailed discussion of the results in the context of the current literature is missing. Further, the discussion of the clinical relevance of the findings is too short and incomplete. In addition, there is no correlation to the mentioned patients' data. Since the authors refer to patients carrying mutations in SLC44A2, a comparison of the human disease manifestations to the described knock-out mouse would be mandatory.
3. Since the authors mention a correlation between deep vein thrombosis and variants in SLC44A2, exploring the phenotype of Slc44a2^{-/-} mice, especially in VTE experiments is essential.
4. As shown in supplemental table 1, the full knock-out of Slc44a2 dramatically alters WBC and lymphocyte counts, which is not addressed in this manuscript. How do the reduced counts affect DVT or in vivo thrombosis?
5. The performed BMC transfer experiments indicate a role of Slc44a2-deficient hematopoietic cells in the regulation of hemostasis. However, this manuscript does not provide evidence for a role of platelet-derived Slc44a2 in hemostasis. Were platelet transfusion experiments performed to test whether the observed phenotype is platelet-dependent?
6. The authors show results from in vivo thrombosis models: FeCl₃-induced injury of the A. carotis and of mesenteric arterioles. Both models use FeCl₃ as a chemical agent to induce endothelial injury and therefore rely on exposed tissue factor to a great extent. Mechanical injuries rather lead to the exposure of collagen with a lower overall contribution of tissue factor. Do Slc44a2 mice display a comparable prolongation of thrombus formation in such models?
7. The here displayed platelet activation and aggregation studies were only performed after thrombin stimulation. Do Slc44a2^{-/-} platelets present with similar activation defects when stimulated with (hem)ITAM-coupled or other agonists? Were different thrombin concentrations tested for the different assays? Can the observed defects be overcome by increasing thrombin concentrations?
8. Were alterations in the granule content observed? How are dense granules and mitochondria linked? Are granule numbers also altered?
9. Statistics: Please clarify your significances: Which groups are compared to each other (e.g. Figure 2c: Asterisks on ADP-stimulated samples (WT and KO) – compared to which sample?)

Response to Reviewers.

We thank our reviewers for your suggestions and constructive criticisms. We have tried as best we can to address the many points you raised. Our data and conclusions are stronger as a result of your ideas.

Reviewers' comments:

Reviewer #1 (Remarks to the Author):

General comments:

This is a generally well-written manuscript in which the authors, through a set of cleverly designed experiments, demonstrate that the choline transporter Slc44a2 controls platelet activation through a mechanism involving mitochondrial production of ATP. This study unravels a new regulator of platelet activation that might be eventually used for the modulation of diseases involving an abnormal platelet aggregation. To demonstrate their hypothesis, Bennet et al. take advantage of the use of a Slc44a2 knockout mice, that has been demonstrated to have impaired platelet activation. We highlight the number of different techniques used in the study, from molecular biology, to sub-cellular fractioning or mass spectrometry. The order of experiments flows reasonably well, allowing the reader to follow the paper easily. Therefore, many of the possible questions that arise from a set of results are readily solved in the next figure. However, the presentation of the data is poor and somehow confusing, not matching its quality. This is an issue that should be improved in the next versions of the manuscript.

This work provides a well-designed approach for the characterization of a molecular target of platelet function beyond the thrombotic pathway, as well as, an endogenous metabolite (ADP) that can modulate platelet activation when it is added exogenously. All in all, we do not think that there is any major concern that precludes the publication of this article. However, several minor concerns should be adequately addressed before its publication.

Author response: Thank you for comments. We agree that our data point towards a novel regulator of platelet activation with therapeutic potential. We've used complementary approaches (molecular biology, cellular fractionation, and mass spectrometry) to study this pathway. We've generated a lot of data, and we appreciate your recommendations for us to clarify the presentation of our data.

Minor concerns:

1. Presentation of data should be improved in a global way. Resolution of figures is low, colors are sometimes difficult to distinguish, and doses of some molecules are missed. We encourage the authors to improve the quality of figures, facilitating readers to follow them more easily. Also, captions should contain more information. In particular, the next points should be addressed:
 - For all the drugs/inhibitors added, the dose should be detailed either in the graph or in the caption. This applies at least for figures 2a, 2c, 2f-h, 3-h-j, 4d, 4f-g.
 - Captions information should be enough to minimally understand the experiment. Please, avoid descriptions like "Platelet GPIIb/IIIa activation" (2d) and similar.
 - Figure 1c→ The WB quality is very low. It seems that you have the spots for the first and third condition and a blank hole in the middle. Please, **repeat the WB** or show a better image. Also, explain why HEK293 are used as control in the text or caption.
 - Figure 1d→ There is redundant information (legend and X-axis showing the same information).

- Figure 1e→ Explain in the text (last paragraph of page2) that you are transplanting bone marrow cells between mice.
- Figure 1f→ Upper panel is impossible to read. Make it bigger.
- Figure 1g→ Time unit is shown as “sec” when in panel d is shown as seconds. Please unify, or even better, show international system notation (s).
- Figure 1f→ X-axis labeling is not aligned with the bars. Rotate it vertically.
- Figure 1a→ Impossible to see anything.
- Figure 2g-h→ Some bars are thicker than others. Very confusing to follow in general. Please, use colors to help with the identification of which bar corresponds to which treatment.
- Figure 3e-f→ If the magnitude measured is supposed to be the same (choline transport), why the Y-axis units are different? Please, clarify this. In order to distinguish between the 2 panels, indicate that the second one is isolated mitochondria (even when it is expressed in the caption).
- Figure 3g→ Use different colors for dots that you consider of interest for the project instead of using arrows. Show the legend with the color of every metabolite. Make it bigger.
- Figure 4g→ Use colors for the bars. The second and third one can be barely distinguished. I do not know if that grouping is the best way to make your data more impactful. I encourage you to figure out another way to make that graph with 3 variables (WT/KO, ADP and Thrombin) easier to follow.

Author response: Thank you for comments about Figure 1. We have improved the presentation of all panels in Figure 1-4. We have repeated the Western blot in Figure 1, and the new Western blot shows the same thing as the original Western blot: wild-type mice express Slc44a2, and knockout mice do not. Please note that the blank lane in the middle of the gel is the lane for the knockout mice that do not express the protein. We also include negative controls (HEK293 cells transfected with an empty vector) and positive controls (HEK293 cells transfected with Slc44a2 vector) so you can confirm that we are identifying the correct protein.

We have added the following text to the paragraph describing Figure 1c. “Relative expression of Slc44a2 is higher in the heart than in most other tissues for reasons that are unknown. We included two controls for immunoblotting, lysates of HEK293 cells transfected with an empty expression vector in which Slc44a2 is absent, and lysates of HEK2903 cells transfected with a Slc44a2 expression vector, in which Slc44a2 is present (Fig. 1C). Mice lacking Slc44a2, Slc44a2(KO), mice, lack Slc44a2 expression in their platelets and in their bone marrow (Fig. 1c).”

2. A scheme with the proposed working model should be included as new figure 5. Since the data presented in this manuscript supports a complex model of platelet activation through a tightly regulated mitochondrial function, the inclusion of a figure with the model proposed for Slc44a2 will clearly help readers to have a global idea without paying much attention to the results.

Author response: Thank you for your good idea about presenting a working model. We've taken your suggestion and create a new summary figure panel and added it to Figure 5

3. In figure 1a, we missed the expression of Slc44a2 in bone marrow cells. Especially, because bone marrow cell transplants are carried out in other experiments. Please include it. We would

also like to have an explanation of why Slc44a2 expression is that high in heart. Could it be because β -actin is lower in heart? Do you think that using β -actin as housekeeping was the best choice for this experiment?

Author response: Thank you for your question about expression of Slc44a2 in bone marrow cells. We have performed a new experiment, a Western blot for Slc44a2 in bone marrow cells from wild type and from knockout mice, and we added it to the Western blot in Figure 1. We don't know why Slc44a2 is expressed in different levels in different organs; beta-actin is used by many groups as an internal house-keeping protein for many studies.

We have added the following text to the paragraph describing Figure 1c. "Relative expression of Slc44a2 is higher in the heart than in most other tissues for reasons that are unknown. We included two controls for immunoblotting, lysates of HEK293 cells transfected with an empty expression vector in which Slc44a2 is absent, and lysates of HEK2903 cells transfected with a Slc44a2 expression vector, in which Slc44a2 is present (Fig. 1C). Mice lacking Slc44a2, Slc44a2(KO), mice, lack Slc44a2 expression in their platelets and in their bone marrow (Fig. 1c)."

4. In figure 2c, different agonists are used to stimulate platelets, observing that ADP has an impact in KO mice. Have you included any reference that explains why ADP is pivotal for platelet activation? Otherwise, it seems that the fact of including ADP was a random choice.

In the line 10 of page 8, it is stated that "adding exogenous ADP should rescue the platelet activation decrease". In that particular form it seems that the idea of adding ADP arises as a consequence of the previous results (which is true in part). But also, it is perfectly known that ADP is a key player in platelet activation. Please include this fact at any point of the text. We also request the authors to show the effect of an analogous non-hydrolysable molecule of ATP in platelet activation, since the rationale to add ADP is the same to add ATP.

Author response: Thank you for your question about ADP as an agonist for platelet activation. You are right, adding ADP to rescue the defect was a deliberate choice (not a random choice), and we have added a sentence and a reference in the manuscript explaining that ADP is a major endogenous agonist that activates platelets. There are several classes of agonists that activate platelets through receptors. For example, thrombin activates the family of PAR receptors, ADP activates platelets through the purinergic receptor P2Y12. ADP is a major platelet activator that is made by platelets and released by platelets to stimulate neighboring platelets by autocrine and paracrine pathways. The central point of our study is that platelets that cannot metabolize choline cannot make ATP and ADP, and cannot activate themselves. We demonstrate this concept by a series of studies. In one of these studies, we show that platelets that cannot metabolize choline have decreased ADP levels, but adding back ADP restores activation. (Please note that ATP is not a major activator of platelets. We added a new experiment as you suggested, adding on hydrolysable ATP to platelets, and as expected, non-hydrolysable ATP increases platelet activation by a small amount, but does not fully rescue the defect in ADP.)

We have added the following sentence to the paragraph in the Results section describing Figure 2:

We compared the response of Slc44a2(WT) and Slc44a2(KO) platelets to various agonists, such as thrombin which signals through G protein coupled receptors, and convulxin which interacts with glycoprotein VI (GPVI) and signals through a (hem)immunoreceptor tyrosine-based activation motif (ITAM)-dependent pathway [Ref 11,12].

We have added the following sentences and references to the paragraph in the Discussion section:

Platelets secrete ADP after stimulation by multiple extracellular agonists such as thrombin [Ref 16]. After its release, ADP then amplifies platelet activation in an autocrine and paracrine manner through the platelet P2Y12 receptor [Ref 17].

5. In the second line of page 6, please replace “spectrometric” by “spectrometry”.

Author response: Thank you for your corrections.

6. In the first paragraph of page 6, it is stated that “Mice lacking Slc44a2 have decreased choline and its metabolites such as betaine and dimethylglycine”. Observing the figure 3g, it seems that betaine change is not statistically significant. With our interpretation of the Y-axis unit, when $-\log P < 1$, the pvalue is greater than 0.1, so betaine is not (by far) statistically significant. Is this correct? If so, please modify the text. Actually, I would say that betaine is not changed at all (fold change of 0.95 cannot be considered relevant).

Author response: We agree. We’ve changed the text to say that dimethylglycine is statistically decreased (and we have eliminated the discussion of betaine).

7. In the second paragraph of page 7, FCCP is not introduced. It should be specified why you are using this compound for your assay.

Author response: Thank you for your suggestion, we have explained the use of FCCP in the text. We added the following sentence:

The maximum rate of ATP production is greater in Slc44a2(WT) platelets than in Slc44a2(KO) platelets, following treatment with the membrane uncoupler FCCP (carbonyl-cyanide-4-(trifluoromethoxy) phenylhydrazone) (Fig. 3h after FCCP).

8. In general, the results are powerful, but their discussion is vague and for some points is even incomplete. For instance, is it possible to find a connection between choline (and phosphatidylcholine phospholipids) metabolism and cardiolipins? Cardiolipins are very important lipids in the inner mitochondrial membrane and they have been demonstrated as key molecules for the oxidative phosphorylation. Also, patients with anti-cardiolipin antibodies can present thrombotic events.

Author response: Yes we agree that the discussion is incomplete. This is due to the severe space limitations of this particular journal. We thank you for your ideas. We have modified the discussion, and we have added a section as you propose, suggesting that choline might affect mitochondrial oxidative phosphorylation through effects on mitochondrial membranes. We have added the following sentence to the discussion:

Another possibility is that Slc44a2 mediates choline transport into mitochondria where it is metabolized into structural compounds such as phosphatidylcholine phospholipids that stabilize mitochondrial membranes which are necessary for oxidative phosphorylation [Ref 13, 14, 18].

Reviewer #2 (Remarks to the Author):

This is an interesting and in general well-preformed study. Overall, the data is of high quality and supportive of the overall conclusions. I have a few suggestions the authors might wish to consider.

The authors state that 'Taken together, our data suggest that Slc44a2 control mitochondrial production of ROS during platelet activation'. This conclusion is based I believe on the data Figure 2g where the KO platelets do not produce an increase in ROS levels following thrombin stimulation. However, I would caution the authors from this strong of a conclusion since the KO cells do not respond at all to thrombin (i.e. no P-selectin, fibrinogen, etc.). As such, it is unclear whether there is a direct effect of Slc44a2 on ROS levels or a more general effect on thrombin-mediated activation. Looking at mitochondrial ROS levels in response to ADP might resolve this issue since Slc44a2 KO platelets respond to this stimulus.

Author response: Thank you for suggestion. We have performed the experiment you recommend. We added ADP to wild-type and knockout platelets, and show that mitochondria from Slc44a2(KO) cells fail to respond to ADP (new Figure 4h-i). These data suggest that there is an intrinsic defect of mitochondria in Slc44a2(KO) cells. These data support the idea that there is a direct effect of Slc44a2 on mitochondrial ROS production. And these data support our hypothesis that Slc44a2 transports choline into mitochondria where it is metabolized into substrates that support the production of ATP and ROS.

Secondly, while the overall data quality is high, I have two issues the authors can hopefully resolve.

First, the change in ATP and ADP in Fig 3g is not significant while in Figure 4 it is. Is there a reasonable explanation for this?

Author response: Thank you for comments. Yes, you are right, the change in ATP and ADP between wild-type and knockout platelets is significant in Figure 4 but not in Figure 3. We think this is due to the measurement techniques: mass spectrometry measurement of levels ADP and ATP might not be as sensitive as luciferase measurements. Another possibility is that the preparation of the samples for mass spectrometry and luciferase assays contributes to the difference.

Secondly, the response to choline in Fig 3j looks quite modest. Is this significant/reproducible? Finally, the data in Figure 3g is of interest and if possible should be shown in a more usable form as a Supplemental Table as well as in its current volcano plot.

Author response: Thank you for your observations. When we add choline to the mitochondria, oxygen consumption rates jump by 40%. Pyruvate is the major fuel of mitochondria, and we propose that choline is an additional fuel but not the major fuel. This boost in oxygen consumption is reproducible, and we repeated this experiment again, and now show the new data in this new figure 3i.

At your recommendation, we have now placed the metabolomics data in Fig 3g into a Supplemental Table provided as an excel file.

Reviewer #3 (Remarks to the Author):

In this study entitled, "The choline transporter Slc44a2 controls platelet activation and thrombosis by regulating mitochondrial function", Bennett and colleagues propose the choline transporter Slc44a2 as a novel therapeutic target based on the observation that Slc44a2-deficient mice display a prolonged time to in vivo thrombus formation. Furthermore, the authors link mitochondrial metabolism to platelet activation.

In general, the data shown in this manuscript is interesting but at the present stage considered preliminary. Further, the conclusions are not always fully supported by the presented data. I have the following major concerns:

1. An Abstract summarizing the main findings of the manuscript is missing.

Author response: The Abstract is the first paragraph. We will label the first paragraph appropriately. Thank you.

2. A detailed discussion of the results in the context of the current literature is missing. Further, the discussion of the clinical relevance of the findings is too short and incomplete. In addition, there is no correlation to the mentioned patients' data. Since the authors refer to patients carrying mutations in SLC44A2, a comparison of the human disease manifestations to the described knock-out mouse would be mandatory.

Author response: We agree, the discussion is too short. This is due to the space limitations imposed by the journal (5000 word maximum). We added to the discussion a paragraph discussing the human genetic data. We have added an extra paragraph to the discussion section to address the imitations of our study. Please note that adding this text greatly increases the length of the manuscript above the 5000 word maximum limit imposed by the journal. We added the following text to the Discussion section:

GWAS of human subjects have shown that genetic variants in the SLC44A2 locus are linked to abnormalities in risk of venous thromboembolism (VTE)^{3,4}. Our data support these findings by demonstrating that Slc44a2 regulates platelet activation and thrombosis in mice (Figs. 1-2). The human GWAS data also show that the genetic variants in the SLC44A2 locus are not associated with known hemostatic biomarkers^{3,4}. These findings support our proposal that Slc44a2 regulates a novel pathway that regulates thrombosis.

3. Since the authors mention a correlation between deep vein thrombosis and variants in SLC44A2, exploring the phenotype of Slc44a2^{-/-} mice, especially in VTE experiments is essential.

Author response: This is an excellent idea. Our original in vivo data show that Slc44a2 regulates tail bleeding and mesenteric thrombosis. We have now performed new experiments, as you suggest, with a murine model of DVT. Our new data show that Slc44a2 controls formation of venous thrombosis in the IVC. We have added these new panels to Figure 1 (new Fig. 1i-k).

4. As shown in supplemental table 1, the full knock-out of Slc44a2 dramatically alters WBC and lymphocyte counts, which is not addressed in this manuscript. How do the reduced counts affect DVT or in vivo thrombosis?

Author response: This is a good point: we do not know why Slc44a2 affects leukocyte counts, and we do not know how the change in leukocytes affects thrombosis. However, we do know that Slc44a2 directly affects platelets from our ex vivo experiments. We have added a comment to the discussion to reflect your concern as follows:

Our study has several limitations. Although we show that Slc44a2 directly affects platelet function (since platelets isolated from Slc44a2(KO) mice have activation defects), other cells lacking Slc44a2 may also contribute to the hemostatic and thrombotic abnormalities in Slc44a2(KO) mice. For example, knockout mice have decreased lymphocyte counts compared to wild-type mice [Supplemental Data], and lymphocytes might regulate thrombosis in vivo. [REFERENCES].

5. The performed BMC transfer experiments indicate a role of Slc44a2-deficient hematopoietic cells in the regulation of hemostasis. However, this manuscript does not provide evidence for are role of platelet-derived Slc44a2 in hemostasis. Were platelet transfusion experiments performed to test whether the observed phenotype is platelet-dependent?

Authors response: yes you are correct, we show that knockout mice have thrombosis defects and we show that knockout platelets ex vivo have activation defects, but we do not show that platelets inside the knockout mice are the major cause of the thrombosis defects. We have added this fact as a limitation to our study in the Discussion. We will address cell specific effects of Slc4a2 in future studies.

6. The authors show results from in vivo thrombosis models: FeCl₃-induced injury of the A. carotids and of mesenteric arterioles. Both models use FeCl₃ as a chemical agent to induce endothelial injury and therefore rely on exposed tissue factor to a great extent. Mechanical injuries rather lead to the exposure of collagen with a lower overall contribution of tissue factor. Do Slc44a2 mice display a comparable prolongation of thrombus formation in such models?

Authors response: this is an excellent point. The question is: do Slc44a2(KO) mice have defects in thrombosis when stimulated by different triggers, some involving tissue factor, others involving mechanical injuries, To address this question, we added a new thrombosis model, one using mechanical injury of the IVC to cause DVT. In this new model, Slc44a2 display delayed thrombus formation. We have added this new murine model of DVT to Figure 1 (new Fig. 1i-k).

7. The here displayed platelet activation and aggregation studies were only performed after thrombin stimulation. Do Slc44a2^{-/-} platelets present with similar activation defects when stimulated with(hem)ITAM-coupled or other agonists? Were different thrombin concentrations tested for the different assays? Can the observed defects be overcome by increasing thrombin concentrations?

Authors response: you raise important questions. We have taken your advice and performed several new experiments. First we performed dose response studies with thrombin activating wild type and knockout platelets. As you can see, knockout platelets have defective responses to thrombin even at high levels of thrombin (new Figure 2e and 2g). Then we stimulated platelets with agonists targeting diverse receptors (thrombin signaling through PAR, ADP signaling through P2Y₁₂, thromboxane analogs through IP₁, and convulxin signaling through GPIIb/IIIa/hem/ITAM. Platelets from knockout mice have

activation defects when stimulated by all these agonists, except for ADP P2Y12 signaling (new Fig. 2f).

8. Were alterations in the granule content observed? How are dense granules and mitochondria linked? Are granule numbers also altered?

Author response. This is an interesting concept. We counted alpha granules and dense granules in platelets from wild type and knockout mice in electron microscope photomicrographs. In platelets with Slc44a2 and without Slc44a2, there are similar numbers of alpha and dense granules (new Supplemental Figure 1).

9. Statistics: Please clarify your significances: Which groups are compared to each other (e.g. Figure 2c: Asterisks on ADP-stimulated samples (WT and KO) – compared to which sample?)

Author response: I apologize we were not more clear in our statistics. We checked our other figures as well. We usually compare wild-type to knockout platelets. For example, we compare unstimulated wild-type and unstimulated knockout platelets. Then we compare stimulated wild-type platelets to stimulated knockout platelets.

Reviewers' Comments:

Reviewer #1:

Remarks to the Author:

The authors have satisfactorily addressed the concerns.

Reviewer #2:

Remarks to the Author:

The authors have been very responsive and have completely addressed my previous concerns

Reviewer #3:

Remarks to the Author:

Re-Review on Bennett et al.

The authors addressed some of the reviewer's questions and added new data to the manuscript. The revised version now comprises a deeper analysis on Slc44a2-mediated mitochondrial metabolism, however, several questions still remain and a few more came up regarding the new experiments, thus making the manuscript still improper for publication.

Line 69: How were the mice generated? Are they conditional KOs only lacking Slc44a2 in BM cells?
Fig 1 c: GAPDH line looks fully oversaturated and artificial.

Fig 1 f: Representative images showing in vivo thrombus formation on FeCl₃-injured mesenteric arterioles do not fit the quantification. While there is almost no thrombus formation in the KO mouse detectable after 20 min in the representative image, the bar graph shows a mean occlusion time of 1100 s. How many mice were not able to form an occlusive thrombus (as the images would suggest) within the observation period of 30 min? The data may be displayed as dots with one dot representing one arteriole/animal. Did the WT mice really occlude after 200 s? If so, why is this not evident in the representative images?

Fig 1 g/h: Similar comment as for Fig 1f: the blood flow chart (esp. for Ko -> Ko and Ko -> WT) does not fit to the quantification in h. The standard deviation is very high in h. How many vessels did not occlude within the observation period? You can't display your results in a bar graph if several mice did not occlude.

Fig 1 e-h: Authors did not include platelets transfusion experiments, which would specify whether the observed hemostasis and thrombosis defect is due to altered platelet activation or reduced WBC counts (almost 50% reduction in KO mice!).

Fig 1 i: Thank you for including results on DVT in Slc44a2-deficient mice, which are in line with the proposed GWAS studies. A discussion of the contribution of immune cells, most importantly neutrophils, to DVT is missing, especially with regard to the altered general blood parameters in Slc44a2-deficient mice. How is the function of other immune cells altered influencing thrombosis and hemostasis?

The differences of the used venous/arterial thrombosis models is not taken into consideration in the discussion of the data:

- DVT model: venous vascular bed, involvement of lymphocytes (reduces amount in KO mice), stenosis model over longer time period (here 6 h, usually 24-48 h), "red thrombus"

- Arterial thrombosis model: no involvement of neutrophils, no NET formation, primarily composed of platelets -> thrombus composition "white thrombus"

- o Mechanical injury not comparable to stenosis, mechanical model missing

Line 121: Please specify in the text: Slc44a2(KO) platelets also have less activation of GPIIb/IIIa than Slc44a2(WT)122 platelets (Fig. 2g). KO platelets show diminished integrin α IIB β 3 activation upon stimulation with which agonists?

Fig 2a-d: Here, platelet activation is measured and not aggregation! A flow cytometric analysis not

gated on platelet activation markers does not allow any conclusion on platelet aggregation, which is typically measured using light transmission or impedance. Besides this, the gating strategy remains unclear due to missing labeling of the y-axis in Fig. 2a. Strong platelet activation as achieved with the used thrombin concentration in Fig 2b normally results in over 90% double positive (Fibrinogen/CD62/CD63) WT-platelets. However, ADP as weak platelet agonist does not result in comparable platelet activation as thrombin.

Fig 2d: Why is the aggregation response towards ADP not altered, ROS production on the other hand though seems to be impaired (Fig 4i). Can you then really conclude that reduced ROS generation results in impaired activation of platelets?

Fig 2 e -f: How do you explain the variable resting levels for untreated WT controls? How do you explain the different P-selectin exposure upon stimulation with 0.5 u/ml thrombin in WT platelets in 2e and 2f – this is a 2x fold difference!

Fig 2k: It still remains unclear whether reduced ROS production is cause or consequence of altered platelet activation. Therefore, also the conclusion in line 138 f. needs to be attenuated.

Fig 3b: Were mitochondria per platelet counted? It is not possible to display mitochondria per EM field since the number of platelets might differ as in Fig 3b.

Fig 3d: What is exactly displayed on the left blot? Mitochondria? Fractioned mitochondria? Platelets? Please specify.

Fig 3i: difference in OCR at 0 min -> how does this account for differences after stimulation?

Fig 4 c/d: Taken into account that the overall ADP content is reduced in KO platelets, is the normalized release still reduced or is it simply due to reduced amounts of ADP in dense granules?

Fig 4g: Resting levels of P-selectin exposure are again not comparable at all to the ones shown in Fig 2 e-f? How can the results differ that much?

Fig 4h-i: Similarly to P-selectin exposure, why do the oxidation values in vehicle-treated mitochondria differ that much from the ones shown in Fig. 2i?

Line 276: Conclusion not addressed by experiments.

Minor:

Data presentation overall is still very poor. Differing fonts and font sized, bad labelling, difficult to understand legends.

Fig 1e: label is missing

Fig 2f: y-axes missing from 700 onwards.

Fig 3i: x-axes labeling: 0 is somewhere, but not at the axis.

Fig 5: The model is too basic to be considered useful. The effect of ROS is not mentioned at all, which would make sense since the authors propose decreased ROS production to account for the observed phenotype.

Response to Reviewers.

We thank our reviewers for your suggestions and constructive criticisms. We have tried as best we can to address the many points you raised. Our data and conclusions are stronger as a result of your ideas.

Reviewers' comments:

Reviewer #1 (Remarks to the Author):

The authors have satisfactorily addressed the concerns.

Reviewer #2 (Remarks to the Author):

The authors have been very responsive and have completely addressed my previous concerns

Reviewer #3 (Remarks to the Author):

Re-Review on Bennett et al.

The authors addressed some of the reviewer's questions and added new data to the manuscript. The revised version now comprises a deeper analysis on Slc44a2-mediated mitochondrial metabolism, however, several questions still remain and a few more came up regarding the new experiments, thus making the manuscript still improper for publication.

Reviewer #3: Line 69: How were the mice generated? Are they conditional KOs only lacking Slc44a2 in BM cells?

Author response: Thank you for pointing this out: our description was not clear. We have added more details about how our collaborator generated these mice in the Methods section. We have changed this line 69 in the text to make it clear that these knockout mice lack Slc44a2 in all tissues including bone marrow and platelets:

“Mice lacking Slc44a2, designated as Slc44a2(KO) mice, are global Slc44a2 null mice that lack Slc44a2 expression in all organs including platelets and bone marrow (Fig. 1c) {ref 8}.”

Reviewer #3: Fig 1 c: GAPDH line looks fully oversaturated and artificial.

Author response: you are right, the GAPDH lanes were overloaded with protein. We repeated the entire western blot with all new samples from new mice (new Fig. 1b). This blot clearly shows that Slc44a2(KO) mice lack Slc44a2 in bone marrow cells and in platelets.

Reviewer #3: Fig 1 f: Representative images showing in vivo thrombus formation on FeCl₃-injured mesenteric arterioles do not fit the quantification. While there is almost no thrombus formation in the KO mouse detectable after 20 min in the representative image, the bar graph shows a mean occlusion time of 1100 s. How many mice were not able to form an occlusive thrombus (as the images would suggest) within the observation period of 30 min? The data may

be displayed as dots with one dot representing one arteriole/animal. Did the WT mice really occlude after 200 s? If so, why is this not evident in the representative images?

Author response: you are right, the original bar graph was not clear. We have taken your advice and we replotted the graph, so now we display each individual mouse as an individual point, not the mean as a bar graph. Please note that we stop all bleeding experiments after 20 minutes, so if the mouse continues to bleed up to 20 min, then the bleeding time is scored as 20 min or 1200 sec. We have included this protocol in the methods section.

Yes, the WT mice occlude after 200 sec. The images of the mesenteric arteriole in WT mice show large increases in platelet deposition, as indicated by greatly increased white signal; and we look at the videos to determine when the mass of white platelets stop moving to indicate occlusion.

Reviewer #3: Fig 1 g/h: Similar comment as for Fig 1f: the blood flow chart (esp. for Ko -> Ko and Ko -> WT) does not fit to the quantification in h. The standard deviation is very high in h. How many vessels did not occlude within the observation period? You can't display your results in a bar graph if several mice did not occlude.

Author response: yes, we agree with you, you are right, the original bar graph was not clear. We have taken your advice and we now display each individual mouse as an individual point, not the mean as a bar graph. Please note that we stop all carotid thrombosis at 20 minutes, so if the mouse does not have complete occlusion then the time to cessation of flow is scored as 20 min or 1200 sec.

Reviewer #3: Fig 1 e-h: Authors did not include platelets transfusion experiments, which would specify whether the observed hemostasis and thrombosis defect is due to altered platelet activation or reduced WBC counts (almost 50% reduction in KO mice!).

Author response: You raise an excellent point. Bone marrow transplantation corrects the hemostasis defect but does not prove that platelets are the key cell expressing Slc44a2 (Fig. 1e, 1g, 1h). We have now performed platelet transfusion experiments as you requested. Transfusion of WT platelets into a KO mouse decreases the bleeding time more than transfusion of KO platelets (new Figure 2 l and 2m). These new data suggest that Slc44a2 in platelets is partly responsible for the bleeding defect in Slc44a2(KO) mice.

Reviewer #3: Fig 1 i: Thank you for including results on DVT in Slc44a2-deficient mice, which are in line with the proposed GWAS studies. A discussion of the contribution of immune cells, most importantly neutrophils, to DVT is missing, especially with regard to the altered general blood parameters in Slc44a2-deficient mice. How is the function of other immune cells altered influencing thrombosis and hemostasis? The differences of the used venous/arterial thrombosis models is not taken into consideration in the discussion of the data:

- DVT model: venous vascular bed, involvement of lymphocytes (reduces amount in KO mice), stenosis model over longer time period (here 6 h, usually 24-48 h), "red thrombus"
- Arterial thrombosis model: no involvement of neutrophils, no NET formation, primarily composed of platelets -> thrombus composition "white thrombus"

Author Response: This is an excellent point. You note that neutrophils play a role in DVT, but the number of neutrophils in wild-type and knockout mice are the same, so neutrophils probably do not contribute to the differences seen in wild-type vs. knockout mice. We have added text to our discussion as you suggested, pointing out that immune cells can alter thrombosis. A complete analysis of every cell type in the knockout mouse is beyond the scope of the current work. However, please note that the experiment you suggested, platelet transfusions, shows that platelets are definitely involved in the bleeding defect, although other cells may also be involved as well (new Figure 2l-m).

Reviewer #3: Mechanical injury not comparable to stenosis, mechanical model missing

Author response: We tried several different new models of arterial mechanical injury as requested, but we were not able to detect thrombosis in our models. Our attempts included:

- forceps crush injury of the carotid artery,
- partial ligation combined with crush injury of the carotid artery,
- cryo-injury combined with crush injury of the carotid artery, and
- forceps crush injury of the mesenteric artery.

These four different approaches did not produce thrombosis in our mouse strain. We do not know why these attempts failed: it could be due to differences in the genetic strains for our mice, compared to the experiments of others.

We have now included in our data four different well established in vivo models of thrombosis in our manuscript. All of our in vivo models show the same thing: mice lacking Slc44a2 have a thrombosis defect.

Reviewer #3: Line 121: Please specify in the text: Slc44a2(KO) platelets also have less activation of GPIIb/IIIa than Slc44a2(WT) platelets (Fig. 2g). KO platelets show diminished integrin α IIb β 3 activation upon stimulation with which agonists?

Author Response: Thank you, we have changed this sentence so it now reads:

“ Slc44a2(KO) platelets also have less activation of GPIIb/IIIa than Slc44a2(WT) platelets when treated with thrombin (Fig. 2g).“

Reviewer #3: Fig 2a-d: **Here, platelet activation is measured and not aggregation!** A flow cytometric analysis not gated on platelet activation markers does not allow any conclusion on platelet aggregation, which is typically measured using light transmission or impedance. Besides this, the gating strategy remains unclear due to missing labeling of the y-axis in Fig. 2a. Strong platelet activation as achieved with the used thrombin concentration in Fig 2b normally results in over 90% double positive (Fibrinogen/CD62/CD63) WT-platelets. However, ADP as weak platelet agonist does not result in comparable platelet activation as thrombin.

Author Response: Thank you for insisting that we perform the traditional method for measuring aggregation, light transmission aggregometry. We have now added more data. We show that platelets from mice lacking Slc44a2 have less aggregation than platelets from wild-type mice (new Figure 2a-d).

Reviewer #3: Fig 2d: Why is the aggregation response towards ADP not altered, ROS production on the other hand though seems to be impaired (Fig 4i). Can you then really conclude that reduced ROS generation results in impaired activation of platelets?

Author response: You raise an interesting question that lies at the heart of our data. We believe that the key pathway that is defective in these knockout mouse platelets is production and release of ATP and ADP. If platelets cannot produce normal levels of ATP and cannot release normal amounts of ATP and ADP, then they do not auto-activate. However, adding back ADP or 2-me-ADP will partially restore platelet function. This is exactly what happens in our rescue experiments when we stimulate with 2-me-ADP (Figure 2f and also Figure 4g). So we think that lack of ROS play a minor role (Supplemental Data) but lack of ATP plays a major role in our Slc44a2(KO) platelets.

Fig 2 e - f: How do you explain the variable resting levels for untreated WT controls? How do you explain the different P-selectin exposure upon stimulation with 0.5 u/ml thrombin in WT platelets in 2e and 2f – this is a 2x fold difference!

Author response: you are right, there are different baseline levels of P-selectin display, and there are different levels of maximal P-selectin display after thrombin treatment. The maximal P-selectin exposure after stimulation ranges from 800 MFI to 1000 MFI to 1500 MFI. Possible explanations for these differences include: experiments are performed with different batches of mice on different days; and we use the same antibody but it is stored over a period of months and natural fading of the fluorophore may occur. Each experiment is internally controlled with the same reagents on the same day. (We prefer not to express data as percent changes.)

Reviewer #3: Fig 2k: It still remains unclear whether reduced ROS production is cause or consequence of altered platelet activation. Therefore, also the conclusion in line 138 f. needs to be attenuated.

Author response: Yes, we agree, reduced ROS production is linked to platelet activation, and do not necessarily cause platelet activation. We have limited our conclusion in the discussion line 138. We now write:

“These studies suggest that mitochondria are the source of ROS which accompany platelet activation. Taken together, our data suggest that Slc44a2 controls mitochondrial production of ROS during platelet activation.”

Reviewer #3: Fig 3b: Were mitochondria per platelet counted? It is not possible to display mitochondria per EM field since the number of platelets might differ as in Fig 3b.

Author Response:

We counted the number of mitochondria per field in 50 fields (Fig 3a-b). Then to control for number of platelets, we measured mitochondrial DNA per 50,000 platelets (Fig. 3c). We have added a sentence to the Methods section explaining this procedure:

“ Mitochondrial DNA copy number was measured in 50,000 platelets by qPCR using a commercial kit (Abcam), and normalized to WT copy number. “

Reviewer #3: Fig 3d: What is exactly displayed on the left blot? Mitochondria? Fractioned mitochondria? Platelets? Please specify.

Author Response:

We wrote a figure legend that was unclear, and thank you for asking us to clarify to our readers. The legend for Figure 2d now reads:

“(d) Slc44a2 is present in lysate of mitochondria purified from platelets (left immunoblot) and Slc44a2 is enriched in the mitochondrial fraction of platelets but not in the cytosolic fraction of platelets (right immunoblot).”

Reviewer #3: Fig 3i: difference in OCR at 0 min -> how does this account for differences after stimulation?

Author Response:

This is an excellent question. There is a difference in the oxygen consumption rate of platelets at baseline due to the lack of Slc44a2 (Fig 3h at time 0). There is also a difference in oxygen consumption rate during various types of stress. The kinds of stress reveal the differences. For example, there is a difference in OCR at baseline time 0 min which means at rest mitochondria are consuming less oxygen and producing less ATP. For example, there is a difference in OCR after FCCP treatment time 40 min, which means the maximal uncoupled rate is different. (Please note that there are no differences in OCR after oligomycin which poisons mitochondria complex I and which only permits oxygen consumption from non-mitochondrial enzymes, and after oligomycin there are no differences between groups.) So there are differences between mitochondria from WT and KO mice at baseline and at stress. In Fig 3i, we show that these differences are partly due to choline, since adding choline to WT platelets increases OCR, but adding choline to KO platelets has no effect.

Reviewer #3: Fig 4 c/d: Taken into account that the overall ADP content is reduced in KO platelets, is the normalized release still reduced or is it simply due to reduced amounts of ADP in dense granules?

Author response: This is an interesting question: for Slc44a2(KO) platelets, is decreased ADP release due to less ADP content or less ADP secretion or both? We used our data from Fig. 4b and Fig. 4d, and we calculated the ADP content of WT and KO platelets, and the ADP release of WT and KO platelets, and we normalized release to content.

ATP content:

WT 27,440

KO 21,117

ATP release

WT 16,938

KO 7,970

Normalized ATP release

WT release/content. $16,938 / 27,440 = 0.61$

KO release/content $7,970 / 21,117 = 0.37$

These calculations suggest that for a given amount of internally stored ATP, a knockout platelet releases less ATP than a wild-type platelet. These calculations support our proposal that

platelets lacking Slc44a2 have a defect in secreting ATP.

Reviewer #3: Fig 4g: Resting levels of P-selectin exposure are again not comparable at all to the ones shown in Fig 2 e-f? How can the results differ that much?

Author response: you are right, there are different baseline levels of P-selectin display, and there are different levels of maximal P-selectin display after thrombin treatment. The maximal P-selectin exposure after stimulation ranges from 800 MFI to 1000 MFI to 1500 MFI. Possible explanations for these differences include: experiments are performed with different batches of mice on different days; and we use the same antibody but it is stored over a period of months and natural fading of the fluorophore may occur; and we use the same lots of thrombin but they are aliquoted and stored in the freezer for different lengths of time. Each experiment is internally controlled with the same reagents on the same day. (We prefer not to express data as percent changes.)

Reviewer #3: Fig 4h-i: Similarly to P-selectin exposure, why do the oxidation values in vehicle-treated mitochondria differ that much from the ones shown in Fig. 2i?

Author response: you are right, there are different baseline levels of DCF-DA oxidation and different levels of stimulated oxidation. Possible explanations for these differences include: experiments are performed with different lots of DCF-DA, and with different sets of mice on different days. Please note that maximal ROS levels are greater after stimulation with thrombin compared to stimulation with ADP. Each experiment is internally controlled with the same reagents on the same day. (We prefer not to express data as percent changes.)

Line 276: Conclusion not addressed by experiments.

Author response: you are right, we have removed this line.

Reviewer #3: Minor:

Data presentation overall is still very poor. Differing fonts and font sized, bad labelling, difficult to understand legends.

Fig 1e: label is missing

Fig 2f: y-axes missing from 700 onwards.

Fig 3i: x-axes labeling: 0 is somewhere, but not at the axis.

Fig 5: The model is too basic to be considered useful. The effect of ROS is not mentioned at all, which would make sense since the authors propose decreased ROS production to account for the observed phenotype.

Author response: you are right, there were problems with the presentation of our figures, and we have fixed the problems you specify. The point of Figure 5 is to show that choline is transported into mitochondria, choline affects mitochondria production of ATP, and thus choline regulates platelet activation through ATP release. We added ROS to our figure 5 as you requested.

Reviewers' Comments:

Reviewer #3:

Remarks to the Author:

The authors addressed most of the reviewer's questions and added new data to the manuscript strengthening the overall quality. Other than the reviewer's comments on mitochondrial metabolism part, the platelet part of the paper could still be improved – data are displayed incorrectly, new data is poorly discussed. There are a lot of careless and small errors.

Comments:

- The authors conclude that KO platelets have a defective ADP/ATP secretion, while on the other hand showing that only alpha-granule release is impaired upon stimulation, whereas dense granule release is unaltered. ADP/ATP are stored in dense granules. How do the authors explain these contradicting findings?

Figure 1:

- 1b: Which proteins are shown? Please label the lanes.
- 1d: y-axis labeling cut of the numbers (1 of 1000s is missing). labeling (WT and KO) is doubled.
- 1d, e,f: Please include the whole measuring time in your axis: 20 min for tail bleeding time, 30 min for thrombus formation and indicate non-occluded vessels as dots above the "measuring-stopped" line.
- 1f: Thank you for providing dots to display the results with each dot representing one animal. Unfortunately, the methods description and the answers to the reviewer comment do not fit with the graph. In the methods, an observation period of 30 min is indicated for in vivo thrombus formation. Here, only 20 min are depicted (1200 s). Besides this, in the response to the reviewers the authors also use the term bleeding time for Fig. 1f. Fig 1f, however, displays in vivo thrombus formation in mesenteric arterioles.
- 1h: Same as for Fig 1f: Please adjust your methods part to 20 min observation period – and not 30 min (if this is, how it was done).
- 1f, h: data is still displayed incorrectly: the authors need to distinguish between occluding and non-occluding vessels. Accordingly, they also need to analyze significances using either two-tailed Student's t-test (for occluding vessels) or Fisher's exact test (for non-occluding vessels compared to occluding vessels) and shouldn't display a mean for all data points (it is statistically not correct to calculate mean values and SD of non-occluding vessels).
- Platelet transfusion experiments are new Fig 1l and 1m (not 2l and 2 m, as stated in the rebuttal letter)
- 1d, l: What is the difference between these two panels? It looks the same for me - bleeding time of WT and Slc44a2 Ko mice. In the text, the authors conclude that transfusion of WT platelets into KO mice partially restores hemostasis, nonetheless they do not show a corresponding statistical comparison between Fig. 1l + m. It would therefore be advisable to merge l + m and compare mean bleeding times between WT and WT to KO as well as KO and KO to KO.
- I would like to encourage the authors to change the order of the figures and display all tail bleeding graphs after one another. Please also change the order in the text correspondingly.

Fig. 3:

- b and c: thank you for including a second measurement for mitochondrial content. It still needs to be highlighted that counting mitochondria per EM field is incorrect. In addition, the y axis labeling in c is wrong.

Response to Reviewers.

We thank our reviewers for your suggestions and constructive criticisms. We have addressed the many points you raised. Our data and organization and conclusions are stronger as a result of your ideas.

Reviewers' comments:

Reviewer #1 (Prior Remarks to the Author):

The authors have satisfactorily addressed the concerns.

Reviewer #2 (Prior Remarks to the Author):

The authors have been very responsive and have completely addressed my previous concerns

Reviewer #3 (Current Remarks to the Author):

The authors addressed most of the reviewer's questions and added new data to the manuscript strengthening the overall quality. Other than the reviewer's comments on mitochondrial metabolism part, the platelet part of the paper could still be improved – data are displayed incorrectly, new data is poorly discussed. There are a lot of careless and small errors.

Reviewer #3: The authors conclude that KO platelets have a defective ADP/ATP secretion, while on the other hand showing that only alpha-granule release is impaired upon stimulation, whereas dense granule release is unaltered. ADP/ATP are stored in dense granules. How do the authors explain these contradicting findings?

Author response: Thank you for asking this question: this is a key point, and I'm sorry I didn't explain it better. The primary defect in these platelets is that they do not make enough ATP so there is less ATP and ADP released from their granules. The secretion of dense granules is fairly normal in knockout platelets (the vesicle membranes fuse with the plasma membrane and expose CD63 in Fig. 2h), but the knockout platelets contain much less ATP and less ADP than normal platelets (Fig. 4a-c) and the knockout platelets release much less ATP and ADP than normal platelets (Fig. 4d). Since less ATP and ADP is released, there is less autocrine and paracrine stimulation of platelets, so there is much less alpha-granule secretion (Fig. 2e-f) and less GPIIb/IIIa activation (Fig. 2g). The rescue experiment proves that lack of ATP and ADP release causes this platelet defect, since adding exogenous ADP restores the ability of platelets to release alpha-granules (Fig. 4g). In summary, there is normal dense granule exocytosis but because of less ATP and ADP content, the ATP and ADP release from the platelets is decreased, so there is less autocrine activation and less alpha-granule release.

In the Abstract we write: Platelets lacking Slc44a2 contain less ATP at rest, release less ATP when activated, and have an activation defect that can be rescued by the addition of exogenous ADP.

In the Discussion we write: Mitochondrial dysfunction leads to decrease ATP content in platelets, decreased ATP release from platelets, and decreased extracellular ADP. Decreased ADP in turn leads to less paracrine and autocrine platelet activation.

Reviewer #3: 1b: Which proteins are shown? Please label the lanes.

Author response: Thank you for pointing this out: we have added labels to the lanes of Fig. 1b.

Reviewer #3: 1d: y-axis labeling cut off the numbers (1 of 1000s is missing). labeling (WT and KO) is doubled.

Author response: Thank you for pointing this out: we have extended the Y-axis to 1200 seconds to include the entire measuring time of 20 minutes, we have adjusted the axis label so the entire range of numbers is visible, and we have removed the repeated labeling of WT and KO for Fig. 1d. Thank you for helping us fix this panel.

Reviewer #3: 1d, e,f: Please include the whole measuring time in your axis: 20 min for tail bleeding time, 30 min for thrombus formation and indicate non-occluded vessels as dots above the “measuring-stopped” line.

Author response: we made an error in the methods: all tail bleeding and all mesenteric thrombosis assays and carotid thrombosis assays are carried out for a maximum time of 20 minutes (not 30 minutes). We have changed the text in the methods. We have extended the Y-axis to 1200 seconds so that the whole measuring time is included in the axis as you recommend. As you recommend, we have added a “measuring stopped” line to these panels in Figure 1. Thank you for helping us fix these panels.

Reviewer #3: 1f: Thank you for providing dots to display the results with each dot representing one animal. Unfortunately, the methods description and the answers to the reviewer comment do not fit with the graph. In the methods, an observation period of 30 min is indicated for in vivo thrombus formation. Here, only 20 min are depicted (1200 s). Besides this, in the response to the reviewers the authors also use the term bleeding time for Fig. 1f. Fig 1f, however, displays in vivo thrombus formation in mesenteric arterioles.

Author response: we made an error in the methods: all tail bleeding and all mesenteric thrombosis assays and all carotid artery thrombosis assays are carried out for a maximum of 20 minutes (not 30 minutes). (Yes, you are correct, the Figure 1 panel on mesenteric arteriole thrombosis is a thrombosis assay, not a bleeding assay, and this is noted in the text of the Results section.) We have changed the text in the methods section as follows:

Mouse mesenteric thrombosis model and carotid thrombosis model.

Thrombosis was measured as previously described^{38,39}. For the mesenteric thrombosis model, mice were anesthetized and platelets labeled with DyLight488-antibody to GPIIb/IIIa. A target area containing mesenteric arterioles (120-150 μm in diameter) was externalized for imaging. The arteriole flow was recorded for 3 min at resting condition. Then 1 mm^2 of Whatman paper saturated with 7.5% FeCl_3 solution was applied to the arteriole for 3 min and the arteriole flow

was continuously recorded for a total of 20 min. The time to form a small thrombus (50-pixel diameter) and to full vessel occlusion were recorded. Recording was terminated at the end of 20 min if no occlusion were observed. For the carotid thrombosis model, mice were sedated with 2.5% isoflurane and maintained anesthetized with 2% isoflurane. The common carotid arteries were exposed for a baseline flow recording using an MA1PRB Perivascular Flowprobe and a TS420 Flowmeter (Transonic Systems). Then 1×2 mm Whatmann paper soaked with 1.5 µl 7.5% FeCl₃ solution was applied to the ventral surface of the carotid upstream of the flowprobe for 3 min. Flow measurement was resumed for a total of 20 min after FeCl₃ wash-off. We define occlusion as the absence of blood flow (0 ml/min) for 3 min.

Reviewer #3: 1h: Same as for Fig 1f: Please adjust your methods part to 20 min observation period – and not 30 min (if this is, how it was done).

Author response: we made an error in the methods: all tail bleeding and all mesenteric thrombosis assays and all carotid artery thrombosis assays are carried out for a maximum of 20 minutes (not 30 minutes). We have changed the text in the methods section as above.

Reviewer #3: 1f, h: data is still displayed incorrectly: the authors need to distinguish between occluding and non-occluding vessels. Accordingly, they also need to analyze significances using either two-tailed Student's t-test (for occluding vessels) or Fisher's exact test (for non-occluding vessels compared to occluding vessels) and shouldn't display a mean for all data points (it is statistically not correct to calculate mean values and SD of non-occluding vessels).

Author response: Yes you are correct: we used the wrong statistical test. We checked with our senior statistician and he said exactly what you said: we should use a different method of analysis for categorical data. We have now analyzed the data for these panels using Fisher's exact test in Figure 1 where appropriate (after moving the old panels to a new order, these panels are now Fig. 1e tail bleeding and Fig. 1k carotid thrombosis). We have also added text into the Methods section to describe this as follows:

Statistics.

Data with a normal distribution were analyzed by two-tailed Student's t-test for comparison of two groups, and by ANOVA to compare means of three or more groups. Statistical significance was defined as $P < 0.05$. To compare the tail bleeding time of two groups (bleeding time for *Slc44a2*(WT) and *Slc44a2*(KO) mice in Fig. 1e), a dichotomous measure of cessation of bleeding within 20 minutes (yes or no) was used and compared in a 2 x 2 contingency table. Because of small sample sizes within some of the cells, the Fisher's Exact Probability test was used to compare the two groups. To compare the thrombosis time of two groups (time for cessation of flow in the carotid artery for WT to KO and KO to KO mice in Fig. 1k), a dichotomous measure of thrombosis within 20 minutes (yes or no) was used and compared in a 2 x 2 contingency table. Because of small sample sizes within some of the cells, the Fisher's Exact Probability test was used to compare the two groups. Tukey's test was used for multiple comparisons (such as differences in bleeding time between transfused hosts in Fig. 1l).

We have also noted the use of Fisher's exact test in the legends for panels of Figure 1 where this test was used.

Reviewer #3: Platelet transfusion experiments are new Fig 1l and 1m (not 2l and 2 m, as stated in the rebuttal letter)

Author response: we made an error in the rebuttal letter. You are right – the platelet transfusion assays are in Fig. 1l. (We merged Fig. 1l and Fig. 1m as you suggested below.)

Reviewer #3: 1d, l: What is the difference between these two panels? It looks the same for me - bleeding time of WT and Slc44a2 Ko mice. In the text, the authors conclude that transfusion of WT platelets into KO mice partially restores hemostasis, nonetheless they do not show a corresponding statistical comparison between Fig. 1l + m. It would therefore be advisable to merge l + m and compare mean bleeding times between WT and WT to KO as well as KO and KO to KO.

Author response: we have performed the platelet transfusion experiments as you requested. Then we merged old Figures 1l and old Figure 1m into new Figure 1l as you suggested. As before, the bleeding times between WT and KO mice are significantly different (Fig. 1l). Now we see that WT platelet transfusion into KO mice shortens the bleeding time more than KO platelet transfusion into KO mice (Fig. 1l). (We also found that WT transfusion partially but not completely restores the bleeding time of KO hosts – probably because the KO hosts still have some remaining KO platelets.). These new data show that the effect of Slc44a2 upon bleeding time is at least partially platelet specific.

Reviewer #3: I would like to encourage the authors to change the order of the figures and display all tail bleeding graphs after one another. Please also change the order in the text correspondingly.

Author response: Thank you for pointing out that our presentation of panels for Figure 1 should be improved. We have now changed the order of the panels for figure 1. We originally started with the phenotype, then the bone marrow transplant studies, then the platelet transfusion studies. In this new order of panels, we first show that mice lacking Slc44a2 have a thrombosis defect, we next show that the bone marrow compartment is the source of the phenotype, and finally we show that platelets are the ultimate source of the problem. (Placing all the bleeding times together is another way to organize the data, but our new organization highlights the bone marrow and the platelet as the source of the bleeding phenotype.)

Reviewer #3: Fig. 3:

Reviewer #3: b and c: thank you for including a second measurement for mitochondrial content. It still needs to be highlighted that counting mitochondria per EM field is incorrect. In addition, the y axis labeling in c is wrong.

Author response: You are right, counting mitochondria per EM is not accurate as a quantitative method, so we removed the graph in Fig 3b.

Author response: In Fig. 3c, we changed the Y-axis label so that it is now: mitochondrial DNA relative copy number.

Reviewers' Comments:

Reviewer #3:

Remarks to the Author:

The authors addressed most of the comments satisfactorily.

Minor comments:

1. In Figure 1d and l the exact same measurements are shown for WT and KO mice, nonetheless the significances differ significantly ($p < 0.05$ in d and $p < 0.001$ in l). Is this due to multiple comparisons?
2. The authors were supposed to merge 1d, i and l into 1 graph in order to have a direct comparison between transfused and non-transfused mice. Since they do not observe differences between WT mice transfused with WT platelets and KO mice transfused with WT platelets, the involvement of other cell types can basically be excluded (and the differences are definitely not due to remaining KO platelets in the KO mice).

Response to Reviewers.

We thank our reviewers for your suggestions and constructive criticisms. We have addressed the many points you raised. Our data and organization and conclusions are stronger as a result of your ideas.

Reviewers' comments:

Reviewer #1 (Prior Remarks to the Author):

The authors have satisfactorily addressed the concerns.

Reviewer #2 (Prior Remarks to the Author):

The authors have been very responsive and have completely addressed my previous concerns

Reviewer #3 (Current Remarks to the Author):

REVIEWERS' COMMENTS:

Reviewer #3 (Remarks to the Author):

The authors addressed most of the comments satisfactorily.

Minor comments:

1. In Figure 1d and l the exact same measurements are shown for WT and KO mice, nonetheless the significances differ significantly ($p < 0.05$ in d and $p < 0.001$ in l). Is this due to multiple comparisons?

Author response:

Yes, you are right, the significances are different in Figure 1d and in Figure 1l because of multiple comparisons. In Figure 1d we compare two groups with normal distribution and use the Student's t-test to analyze the data. In Figure 1l you asked us to combine two experiments into one, and in the new panel we compare 4 groups of normally distributed data, so we use Tukey's range test to analyze the data. Please note we made a clerical error and typed in the wrong Tukey's range test number for the significance, and this has now been corrected in the Figure legend.

2. The authors were supposed to merge 1d, i and l into 1 graph in order to have a direct comparison between transfused and non-transfused mice. Since they do not observe differences between WT mice transfused with WT platelets and KO mice transfused with WT platelets, the involvement of other cell types can basically be excluded (and the differences are definitely not due to remaining KO platelets in the KO mice).

Author response:

We agree with the reviewer. The platelet transfusion experiments requested by the reviewer suggest that platelets are the cells responsible for the bleeding defects in *Slc44a2(KO)* mice. The reviewer had a good idea in the last review and asked us to merge Fig. 1d and Fig. 1l, and we did that for the last revision (current Fig 1l). Please note that we cannot merge the Figure 1i with this group because 1i is a bone marrow transplantation study and 1l is a platelet transfusion study.

We agree with the reviewer's conclusion. Since we do not observe differences between WT mice transplanted with WT bone marrow and KO mice transplanted with WT bone marrow in Fig. 1i, the involvement of host cells can be excluded. And since transfusing WT platelets into KO recipients decreases the bleeding time compared to KO platelets transfused into KO hosts, therefore platelets are the cells responsible for the bleeding defects in *Slc44a2(KO)* mice. This was the reason the reviewer made the excellent suggestion to perform platelet transfusion experiments. We have incorporated the reviewer's suggestion which now strengthens our conclusion.